# Preconditioning of overcast-to-broken cloud transitions by riming in marine cold air outbreaks

Florian Tornow[1,2], Andrew S. Ackerman[2], and Ann M. Fridlind[2]

[1]Center for Climate Systems Research, Earth Institute, Columbia University, New York, USA
[2]NASA Goddard Institute for Space Sciences, NY, USA

**Correspondence:** Florian Tornow, NASA Goddard Institute for Space Studies, 2880 Broadway, New York, NY 10025, USA (florian.tornow@nasa.gov)

**Abstract.** Marine cold air outbreaks (CAOs) commonly form overcast cloud decks that transition into broken cloud fields downwind, dramatically altering the local radiation budget. In this study, we investigate the impact of frozen hydrometeors on these transitions. We focus on a CAO case in the NW Atlantic, the location of the multi-year flight campaign ACTIVATE. We use MERRA-2 reanalysis fields to drive large-eddy simulations with mixed-phase two-moment microphysics in a Lagrangian framework. We find that transitions are triggered by substantial rain (rainwater paths $> 25$ g m$^{-2}$) and only simulations that allow for aerosol depletion result in sustained breakups as observed. Using a range of diagnostic ice nucleating particle concentrations, $N_{inp}$, we find that increasing ice progressively accelerates transitions, thus abbreviating the overcast state. Ice particles affect the cloud-topped boundary layer evolution primarily through riming-related processes prior to substantial rain, leading to (1) reduction in cloud liquid water, (2) early consumption of cloud condensation nuclei, and (3) early and light precipitation cooling and moistening below cloud. We refer to these three effects collectively as *preconditioning by riming*. Greater boundary layer aerosol concentrations available as cloud condensation nuclei (CCN) delay the onset of substantial rain. However, cloud breakup and low-CCN concentration final stages are found to be inevitable in this case owing primarily to liquid water path buildup. An ice-modulated cloud transition speed suggests the possibility of a negative cloud-climate feedback. To address prevailing uncertainties in the model representation of mixed-phase processes, the magnitude of ice formation and riming impacts and, thereby, the strength of an associated negative cloud-climate feedback process, requires further observational evaluation by targeting riming hotspots with in situ imaging probes that allow both for characterization of ice particles and abundance of supercooled droplets.

## 1 Introduction

Planetary boundary layer (PBL) clouds are common over the world's oceans, where they often substantially enhance the reflection of sunlight from otherwise dark ocean surfaces, while less affecting the emission of terrestrial radiation owing to typically modest differences between cloud-top and surface temperatures (e.g. Hartmann et al., 1992; L'Ecuyer et al., 2019). Marine cold air outbreaks (CAOs) are associated with a particular form of PBL cloud. At mid-latitudes, they commonly occur in post-frontal conditions of extratropical synoptic systems during winter and its shoulder seasons (Kolstad et al., 2009; Fletcher et al., 2016). The rapid advection of cold air masses over a relatively warm ocean surface induces unusually large surface heat

fluxes of $O(10^2$-$10^3$ W m$^{-2}$), while the PBL is capped by strong subsidence at rates of $O(10^1$-$10^2$ mm s$^{-1}$) (Papritz et al., 2015; Papritz and Spengler, 2017). Both surface fluxes and subsidence rates exceed conditions in commonly studied PBL systems (e.g., Roberts et al., 2012; Myers and Norris, 2013) by an order of magnitude or more. Liquid and also ice condensate rapidly increase, often initially as cloud streets that lead to near-overcast cloud decks with roll-like structures. Further downwind, the overcast cloud deck generally breaks apart into an open cellular structure (Brümmer, 1999; Pithan et al., 2019). Understanding

the transition from overcast to broken state is crucial to authentically capture observed CAO radiative effects (McCoy et al., 2017) in earth system models and numerical weather predictions. Prior studies have demonstrated deficits in current models to represent CAOs (e.g., Abel et al., 2017; Field et al., 2014), which can result in substantial deviations from satellite-inferred radiative effects (Rémillard and Tselioudis, 2015).

  Transitions from overcast to broken cloud decks have been primarily studied in the context of stratocumulus-to-cumulus

(SCT) or closed-to-open-cell transitions. To better understand controlling mechanisms, in situ observations have been collected (e.g., Albrecht et al., 1995, 2019), satellite-based retrievals compiled (e.g., Sandu et al., 2010; Eastman and Wood, 2016; Mohrmann et al., 2019; Christensen et al., 2020), and high-resolution simulations performed (e.g., Wyant et al., 1997; Wang et al., 2010; Sandu and Stevens, 2011). The classic SCT theory indicates that transitions are governed by progressive PBL deepening and decoupling arising from advection towards warmer waters and subsequently growing surface fluxes (Bretherton

and Wyant, 1997; Sandu and Stevens, 2011). Other factors like downwelling longwave radiation (Sandu and Stevens, 2011) and the intensity of large-scale subsidence (van der Dussen et al., 2016) further modify SCTs. In this study, we address precipitation-induced transitions – a mechanism well-studied for warm cloud transitions from closed to open cells. Precipitation stabilizes the PBL through evaporation of rain and drizzle below cloud, leading to a modulation from stratiform to more convective regime (Paluch and Lenschow, 1991; Stevens et al., 1998). Studies that have considered the interactions between precipitation and

aerosol that serve as cloud condensation nuclei (CCN; e.g., Yamaguchi et al., 2017; Goren et al., 2019) have demonstrated that (1) precipitation formation requires microphysical collision and collection processes between cloud droplets and also raindrops, effectively reducing the number of CCN in the PBL, and (2) reduced CCN distribute the cloud condensate over fewer droplets, accelerating raindrop formation. Together these two effects may lead to a positive feedback that is irreversible in typical meteorological and aerosol scenarios and represent a notable permutation of classic SCT theory. Observations confirm the

(local) relation of precipitation and "ultra-clean" conditions (i.e., low-CCN concentrations) by comparing neighboring closed and open cells (Terai et al., 2014), sampling the same air mass before and after the transition (Sarkar et al., 2020; Eastman and Wood, 2016), or inferring processes via satellite imagery upwind from ground-based aerosol observations (Wood et al., 2017) or aircraft observations (Ahn et al., 2017).

  CAOs typically produce mixed-phase clouds that – however formed – are a particular source of uncertainty in climate

projections (McCoy et al., 2015). To better understand mixed-phase clouds, many studies have simulated relatively quiescent Arctic clouds and explored their response to various meteorological conditions (e.g., Young et al., 2018) or microphysical compositions (e.g., Eirund et al., 2019). Other studies focused on cloud-aerosol interactions to highlight the presence of clouds despite the low-CCN concentration environment in higher latitudes (Stevens et al., 2018) and to show how seeding from ship emissions might affect mixed-phase cloud properties (Possner et al., 2017). Understanding sources and sinks of ice nuclei is

another area of ongoing research (e.g., Solomon et al., 2015). Several field campaigns have obtained measurements during CAOs, such as M-PACE (Shupe et al., 2008) and ACCACIA (Young et al., 2016). In these rapidly evolving mixed-phase clouds that develop high liquid and ice water contents, microphysical processes may potentially be amplified compared with more quiescent conditions. Observational evidence indicates an active riming process in areas of high liquid water content (e.g., Fridlind and Ackerman, 2018) that can also coincide with locally reduced droplet number concentrations (e.g., Huang et al., 2017). In turn, riming – a process that collects droplets and, thus, reduces condensation nuclei by number – has the potential to transition high-CCN concentration states away from a regime that is considered potentially stable (Baker and Charlson, 1990).

This study concerns the role of frozen hydrometeors in CAO closed-to-broken cloud transitions. The reduction in liquid water path that results from the competition for condensate when ice is present could retard a rain-induced transition. On the other hand, simulations of mixed-phase clouds and increased ice number concentrations suggest a more rapid decay of optically thick clouds owing to intensified snowfall that cools and moistens layers below cloud through sublimation or melting and evaporation (e.g., Eirund et al., 2019)[1]. A general increase in precipitation in the presence of ice as found in cumulus clouds (Knight et al., 1974) and across various cloud types (Field and Heymsfield, 2015) could be expected to further support a more rapid breakup. Therefore, we pose the question whether ice could slow or hasten transitions from overcast to broken clouds in CAOs. To address this question, we select a case in the NW Atlantic, the study area of the multi-year campaign ACTIVATE (Aerosol Cloud meTeorology Intercations oVer the western ATlantic Experiment, Sorooshian et al., 2019) that dedicates its resources during wintertime and shoulder seasons to CAOs and should allow evaluation of the generality of results found here. We assume the Lagrangian perspective in large-eddy simulations by using a domain that follows the PBL flow, guided by input from MERRA-2 (Modern-Era Retrospective analysis for Research and Applications, version 2) reanalysis. We diagnose characteristic events for each simulation: the start and end of an overcast state and the onset of substantial precipitation. We then examine how adding ice processes impacts the timing of those events.

## 2 Simulations of a Cold Air Outbreak

In the following, we briefly describe the selected CAO case (Section 2.1), the large-eddy simulation (LES) code (Section 2.2), and the setup of our simulations (Section 2.3).

### 2.1 A Cold Air Outbreak in the NW Atlantic

This study focuses on a CAO that occurred between the 17[th] and 19[th] of March 2008 in the NW Atlantic (Fig. 1a) following the passage of a cold front as part of an eastward travelling low-pressure system. This CAO constitutes a shoulder season event and was selected on the basis of weather-state analysis of satellite imagery (pers. comm. George Tselioudis). Location and timing of this case are favorable as they align with the ongoing ACTIVATE campaign. Historically, shoulder seasons host fewer CAOs compared to winter months (Fletcher et al., 2016). Using the cold air outbreak index $\theta_{skin} - \theta_{800hPa}$, in which

---

[1] Eirund et al. (2019) examined idle Arctic stratocumuli and did not consider (microphysical) depletion of CCN, which is critical for the transition from closed to open cells in our simulations

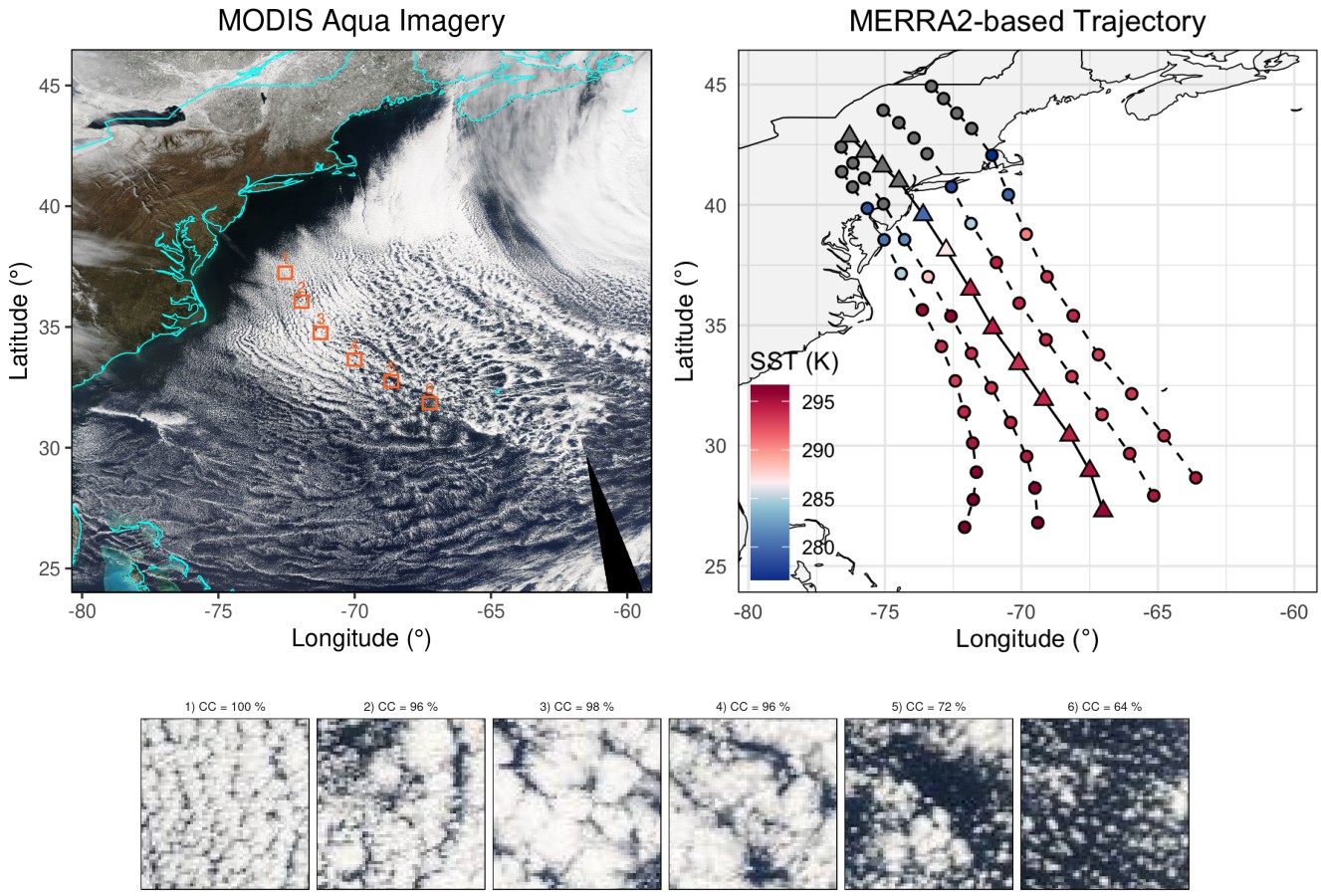

**Figure 1.** (a) Cold air outbreak on March 17, 2008, from MODIS Aqua courtesy NASA Worldview. (b) Near-surface trajectories from MERRA-2, shown in 3-hourly steps, spanning ∼24h (or ∼1500 km): large triangles and the solid line mark the trajectory used for simulations, while circles and dashed lines indicate neighboring trajectories. (c) Detailed $(0.5°)^2$ regions (marked in Fig. 1a). Titles report cloud cover determined from MYD06 data as the portion of $(1 \text{ km})^2$ pixels with a cloud optical thickness greater then or equal 2.5.

$\theta$ is potential temperature, we detected values of up to ~10 K during 17[th] March. This maximum nears the 95[th] percentile of indices collected during winter months in this region (Fletcher et al., 2016).

## 2.2 Eddy-resolving Simulations

We use the DHARMA (Distributed Hydrodynamic Aerosol and Radiative Modelling Application) LES (Ackerman et al., 2004; Zhou et al., 2018) with periodic boundary conditions and a translating coordinate system that follows the PBL mean horizontal
wind. Subgrid fluxes are parametrized using a dynamic Smagorinsky turbulence model (Kirkpatrick et al., 2006). Aerosol available for activation as CCN are represented as a single lognormal mode (geometric mean diameter = 0.05 nm, standard deviation = 1.4, hygroscopicity parameter = 0.55) with a prognostic number concentration. Mixed-phase cloud microphysics is represented based on a two-moment scheme (Morrison et al., 2009), with autoconversion and self-collection following Seifert and Beheng (2001), rain accretion, self-collection, break-up, and fall speed following Seifert (2008), and rain represented as
a gamma size distribution with a shape parameter of 3. Activation of aerosol using a prognostic supersaturation value after microphysical relaxation follows Morrison and Grabowski (2008). Ice formation is treated as in Ovchinnikov et al. (2014) with the mechanistic equivalent of a diagnostic ice nucleating particle number concentration in the immersion mode, $N_{inp}$. Ice nucleation occurs wherever temperatures are below -5° C, supercooled liquid is present, and ice particle number concentrations are below specified $N_{inp}$; here we consider values of 1, 4, and 16 L$^{-1}$. In practice this represents the wide range of ice formation
that might be expected from varying degrees of heterogeneous nucleation and unrepresented ice multiplication, as discussed further below. For simulation without ice we use the shorthand notation "*ice0*". In our simulations, ice is represented as three species: cloud ice, graupel, and snow. When presenting ice mixing ratios, $q_i$, all three categories are summed.

The domain spans 5 km vertically with a uniform spacing of 20 m up to 3.5 km and progressively thicker layers above, using 200 layers in total. The upper 1 km acts as sponge to dampen gravity waves. Horizontally, the domain spans (21.6
km)$^2$ with a horizontal mesh of 150 m. The grid was determined from various combinations (each with similar aspect ratio), ascertaining that the next-higher resolution and also the next-higher domain size result in equivalent PBL evolution of the baseline setup (described further below). To obtain a crude characterization of uncertainty from turbulent noise, we run an ensemble of simulations for the baseline setup of *ice0*, which we effectively assume as representative of other setup variations. Here ensembles are run by varying the seed to the pseudo-random number generator applied to initial fields of water vapor and
potential temperature.

Initial thermodynamic conditions are taken from extracted MERRA-2 profiles (see Section 2.3). Initial aerosol number concentration is set to 200 mg$^{-1}$ below the inversion (roughly guided by $N_d$ derived from MODIS imagery) and to 50 mg$^{-1}$ above (after Abel et al., 2017) in the baseline experiment. We examine sensitivities of this setup in Section 3. We use an aerosol surface source of 70 cm$^{-2}$ s$^{-1}$ (roughly following Clarke et al. 2006 and using the average wind speed of the lowest layer
in extracted profiles). For radiative transfer, we use 385 ppm $CO_2$. From a 30 km-deep profile, we integrate above 5 km to obtain the overlying water vapor (0.004 g cm$^{-2}$) and ozone (5·10$^{17}$ cm$^{-2}$), and we select a representative overlying isothermal layer temperature (130 K) to match the downwelling longwave radiation profile calculated at domain top from the full profile. We nudge horizontal mean temperature and moisture above the main inversion (defined throughout as the mean height of the

maximum vertical gradient of potential temperature) with strength linearly increasing from zero to 500 m above the inversion
and with a time constant of 1 h at full strength. Aerosol number concentrations are nudged to 50 mg$^{-1}$ above the dynamically
defined mean inversion and wind profiles are nudged above 500 m the surface (each with a time constant of 0.5 h). The diurnal
cycle of shortwave radiation is treated using a local time of 4:00 AM at simulation start. We note that reported cloud cover is
computed as the fraction of (150 m)$^2$ columns with optical thickness exceeding 2.5 (treating all hydrometeors as geometric
scatterers where ice optical properties are set as in Fridlind et al., 2012); domain-mean cloud droplet number concentration
($N_c$) is weighted by cloud water to avoid requiring a definition of "in-cloud" (except where indicated otherwise). Table 1
summarizes the setup.

**Table 1.** Baseline model setup.

| Selected aspect | Setup |
| --- | --- |
| LES Dynamics | Stevens et al. (2002) |
| Radiative Transfer | Toon et al. (1989) |
| Surface Similarity | Businger et al. (1971) |
| Subgrid-Scale Mixing | Smagorinsky dynamic turbulence model (Kirkpatrick et al., 2006) |
| Mixed-Phase Cloud Microphysics | Two-moment scheme based on Morrison et al. (2009), extension with raindrop size distribution generalized as gamma distriubtion |
| Autoconversion & Self-Collection | Seifert and Beheng (2001) |
| Rain Accretion, Self-Collection, Breakup, and Fallspeed | Seifert (2008) |
| Prognostic Supersaturation | Morrison and Grabowski (2008) |
| Ice Formation | Ovchinnikov et al. (2014) |
| Horizontal Grid | (21.6 km)$^2$ with 150 m mesh |
| Vertical Grid | 5 km with 20 m mesh from 0-3.5 km and >20 m above |
| Aerosol Size Distribution | lognormal accumulation mode (r$_g$ = 0.05 nm, $\sigma_g$ = 1.4, $\kappa$ =0.55) initally 200 mg$^{-1}$ in the PBL and 50 mg$^{-1}$ in the free troposphere |
| Large-Scale Forcing | MERRA2-based SST and vertical wind |
| Nudging | $<T>$, $<q_v>$, and $<N_a>$ at full strength at 500 m above inversion with $\tau = 1$ h, and $<u>$,$<v>$ at full strength at 500 m above the surface with $\tau = 0.5$ h |

## 2.3  Boundary Conditions from MERRA-2 reanalysis

To drive the LES with a domain that follows the PBL horizontal flow, we extract trajectories from MERRA-2 (Gelaro et al.,
2017) using the horizontal wind at 250 m altitude. To intercept observed roll-like structures (Fig. 1a), we first trace the trajectory
from 33.4° N, 70.1° W, and 1800 UTC forward and backward (amounting to 24 h over ocean surface; Fig. 1b); results also

shown for four neighboring initial points (displaced by intervals of 1° N and 1° E). From MERRA-2 four-dimensional fields (3-hourly fields of ~50 km horizontal resolution and ~72 vertical layers between 1000 and 0 hPa), we extract horizontally interpolated, vertical profiles of meteorological variables (temperature, specific humidity, and wind) per discrete time step. Repeating this extraction with ERA5 fields (1-hourly fields of ~31 km horizontal resolution and ~137 vertical levels; Hersbach et al., 2020), we found similar trajectories (not shown); primarily, profiles of large-scale vertical wind appear smoother in MERRA-2. Sea surface temperature (SST) was also extracted from MERRA-2 and represented as a piecewise-linear evolution along the trajectory with values of 287, 294, and 296.3 K at 0, 3, and 24 h, respectively.

## 3 Results

Satellite imagery of the observed case (Fig. 1a) shows the close succession of cloud streets and rolls that typically form shortly after leaving the continent and the open cellular structures downwind, bounding a relatively narrow region (or short phase in the Lagrangian sense of travelling with the PBL flow) of overcast clouds, here defined as cloud cover above 75 %, as in Christensen et al. (2020). Alternatively, we also consider a cloud cover threshold of 50 % (equivalent to Sandu et al., 2010). MODIS data in Fig. 1c provides an impression of cloud cover. To demonstrate key mechanisms of the transition from overcast to broken cloud field, we first analyze simulations without ice (Section 3.1). By including frozen hydrometeors, we then explore its modification of this transition – in particular the impact of ice on the transition timing (Section 3.2). By changing the levels of $N_{inp}$ (Section 3.3) and initial $N_a$ (Section 3.4) we investigate the robustness of identified impacts to microphysical controls.

### 3.1 Simulation without ice

In this subsection, we demonstrate that the onset of substantial rain (here defined as a rainwater path exceeding 25 g m$^{-2}$) during the overcast phase creates a turning point that leads to the breakup of the cloud deck (i.e., a cloud cover dropping below 75 %). We further emphasize that aerosol removal through microphysical collision processes is a necessary ingredient to cloud breakup, as also found in warm stratocumulus transitions (Yamaguchi et al., 2017).

Figure 2 provides an overview of the evolution of experiments without ice (shown for an ensemble of three simulations) and also with varying $N_{inp}$ that we examine in the following subsections. We show various metrics that characterize the state of the PBL and its condensate (Fig. 2a-2g, 2k, and 2m), variables related to radiative properties (Fig. 2h-2j), the interaction with the free troposphere (FT, Fig. 2l) and changing boundary conditions (Fig. 2n).

Without ice ("ice0"), the PBL rapidly deepens with initial transit from the continent (Fig. 2a), from roughly 2 km at cloud onset to maximum depths greater than 2.5 km, before more gradually becoming shallower after 7–8 h. Prior to the onset of substantial rain (marked by a dot in Fig. 2e, roughly ending a period highlighted by a grey background throughout Fig. 2), supercooled liquid rapidly increases, as the cloud deck fills in after ~1.5 h (Fig. 2b) and thickens reaching a domain mean peak of ~600 g m$^{-2}$ (Fig. 2c). Meanwhile, droplet number concentration decreases monotonically from a peak value only ~1 h after cloud formation (Fig. 2g); as illustrated further below, PBL total aerosol $N_{a+c}$ (activated and unactivated CCN; not shown) gradually diminishes initially from rapid entrainment of FT air with reduced aerosol concentrations and later from

# Cold Air Outbreak Evolution

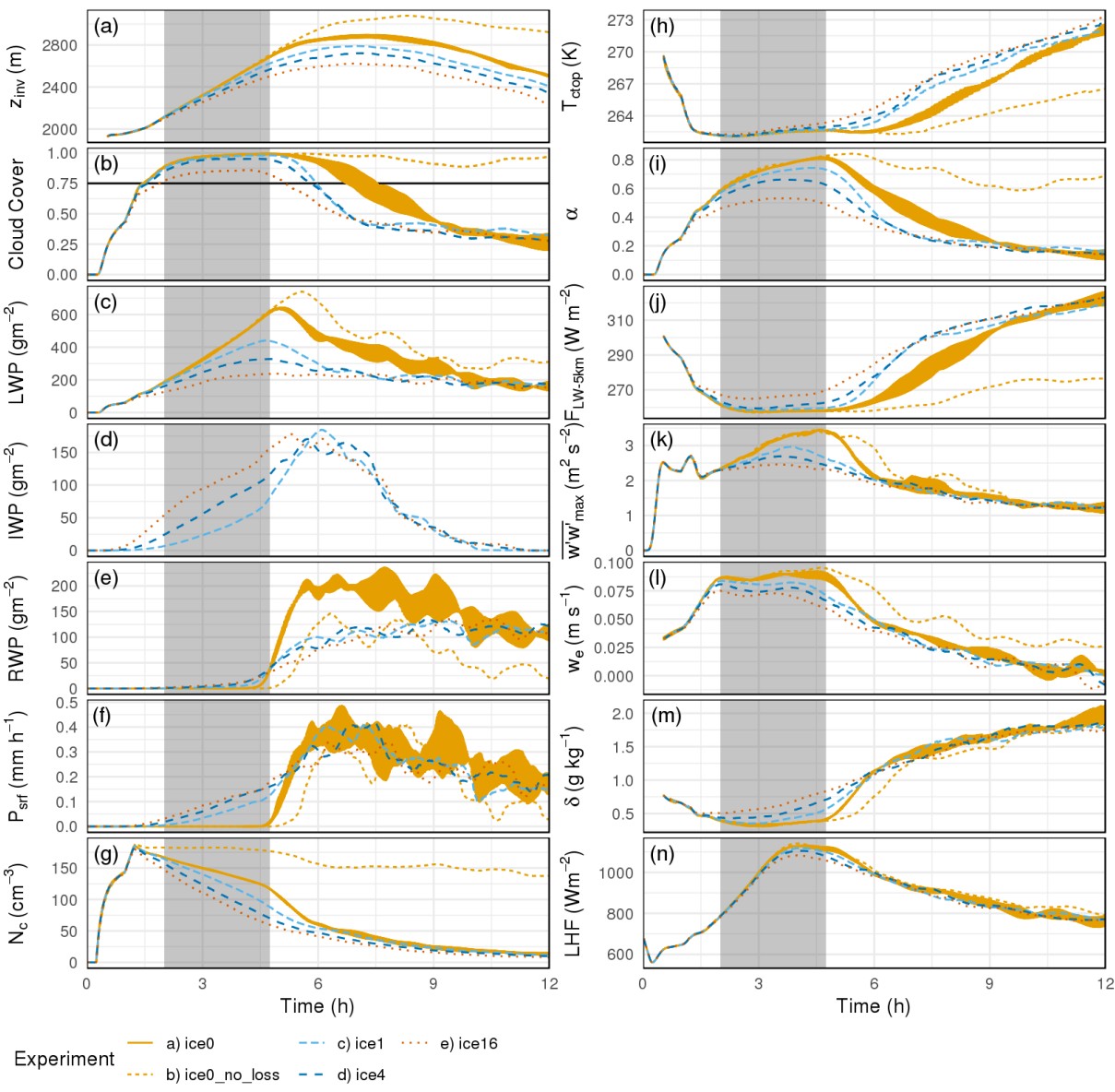

**Figure 2.** Time-evolving (a) inversion height, (b) cloud cover, (c) total liquid water path (including cloud water and rain), (d) ice water path (including cloud ice, graupel, and snow), (e) rain water path, (f) surface precipitation rate, (g) in-cloud droplet number concentration, (h) cloud-top temperature, (i) pseudo-albedo, (j) outgoing longwave radiation at top-of-domain (5 km), (k) domain-maximum column-averaged vertical wind variances, (l) cloud-top entrainment rate, (m) PBL stratification (see text), and (n) surface latent heat flux of four simulations of varying $N_{inp}$ (shown in legend with notation iceN meaning $N_{inp} = N \, \text{L}^{-1}$, and also described in the text). Variables are defined in Sec. 2.2. All values are box-averaged over a lagged 1h-window and domain-mean unless otherwise indicated. Gray areas mark the period introduced in Sec. 3 as "*preconditioning by riming*". For *ice0*, we show the spread over an ensemble of three simulations obtained by changing the pseudo-random seed used in initialization of meteorological fields.

collision-coalescence active in regions of high liquid water mixing ratio $q_c$ (see further details below). Progressively more cloud condensate distributed over fewer droplets initiates substantial rain after $\sim$4.5 h (Fig. 2e). Partial evaporation of rain below cloud contributes to stratification of the PBL (Fig. 2m), reduction of vertical mixing within the PBL (Fig. 2k; depriving the cloud layer of moisture and aerosol from the surface layer) and slowing entrainment (Fig. 2l; which reduces cloud layer drying and aerosol dilution to some degree). After 6 h, the spread within the ensemble becomes noisier due to somewhat stochastic precipitation events.

To examine the overcast-to-broken cloud structural transition, Figure 3 shows cloud geometric extent, rainwater path, and in-cloud vertical motion calculated from three-dimensional domains for hourly snapshots from 4.5 h (the moment of substantial rain onset in all simulations) to 6.5 h (a post-transitional state). In the *ice0* simulation (solid lines in left panels), the cloud vertical structure transitions from a stratiform state at 4.5 h (Fig. 3a, shown in red curve) where most clouds have a geometric thickness of $\sim$1.2 km towards a convective state (shown in green and then blue) where most clouds are geometrically thinner than 1 km and sporadic convection creates few instances of larger vertical extent, as observed in open cells by Wood et al. (2011). The precipitation-induced breakup of overcast cloud deck is generally consistent with findings of Stevens et al. (1998), Savic-Jovcic and Stevens (2008), and Wang and Feingold (2009) in warm clouds and Abel et al. (2017) and Eirund et al. (2019) in mixed-phase clouds.

In contrast to stratocumulus simulations (e.g., Yamaguchi et al., 2017), our results are notably insensitive to increasing the domain size. For a domain size that is four times larger, $(43.2$ km$)^2$, domain-mean time series are scarcely impacted (not shown), and structural statistics also change little (Fig. 3a, dashed versus solid lines). We presume that roll-like structures (shown at rain onset in Fig. 4, top panel) serve as a regular pattern for cloud condensate and rainwater, and this regularity may explain the lack of domain size dependence. In the baseline setup (Fig. 4, bottom panels) roll-like structure are more apparent in the larger domain at 4.5 h. They are preceded by more numerous, smaller cells (at 1 and 2 h) and followed by an assortment of progressively fewer bright and more frequent dim cloud elements (5.5, 6.5 and 9 h), illustrating the commonality of geometrically thinner clouds after the rain onset.

We find a narrowing probability density function (PDF) tail towards sparser occurrences of the largest cloud geometric extent and simultaneously a thickening PDF tail towards increasing occurrences of the greatest rainwater paths in convective elements. This is a result of increased stabilization caused by more buoyant downdrafts above cloud base (per Stevens et al., 1998) and rain evaporation below cloud base, leading to a buildup of moisture in the lowest layers that continue to be fed by large surface fluxes of sensible heat and moisture. Conditional instability in the moistened sub-cloud layer triggers cumuliform convection. When rising, these parcels possess lifting condensation levels of low altitude (not shown). This stabilization of the PBL with increasing precipitation is associated with overall dramatic weakening of in-cloud vertical motion (note logarithmic scales in Fig. 3a, bottom panel). Steadily declining $N_{a+c}$ and slowed vertical motion both additionally promote rain formation (Ovchinnikov et al., 2013) in convective events that then further reduce aerosol concentrations, thus constituting a positive feedback. The cloud deck without ice breaks apart 5-6 h after its formation (Fig. 2b) using our metric (75 % cloud cover). We note that this threshold is somewhat arbitrary; lowering the "overcast" definition to >50 % cloud fraction would correspond to diagnosing longer overcast periods and a greater difference across simulations, as discussed further below.

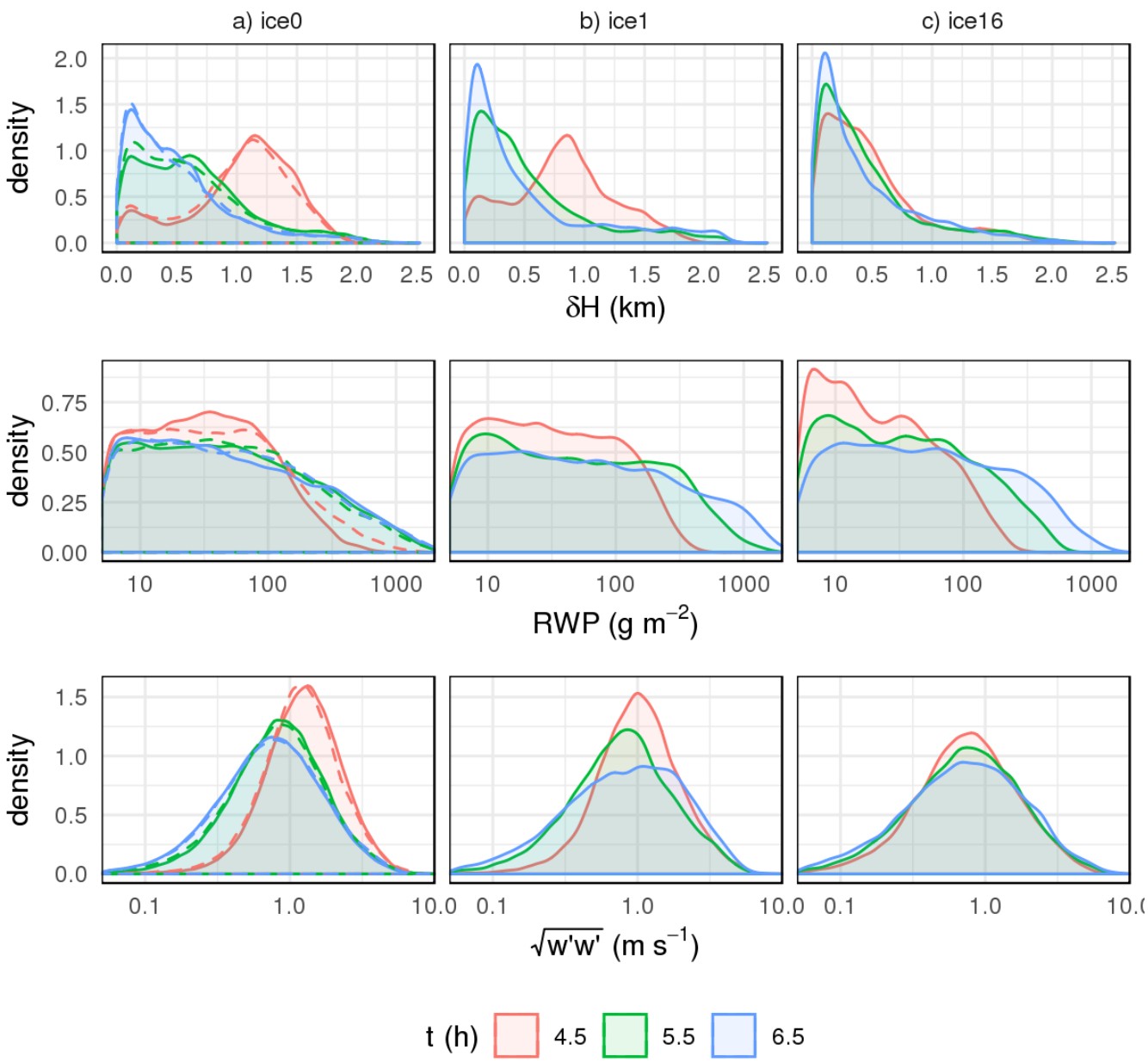

**Figure 3.** Statistics of cloud vertical extent (or geometric thickness), $\delta H = H_{cloud-top} - H_{cloud-base}$, (where $H$ is height above ground) column-average absolute vertical wind speed, $\sqrt{w'w'}$, and rain water path from cloudy columns within 3D domains for three hourly time steps since the onset of precipitation for simulations with (a) *ice0*, (b) $N_{inp} = 1\ \text{L}^{-1}$, and (c) $N_{inp} = 16\ \text{L}^{-1}$. Note logarithmic x-axes for $\delta H$ and $\sqrt{w'w'}$. For *ice0*, we show statistics for an ensemble of three simulations. Dashed lines in the left column refer to a simulation with quadrupled domain horizontal area (see text).

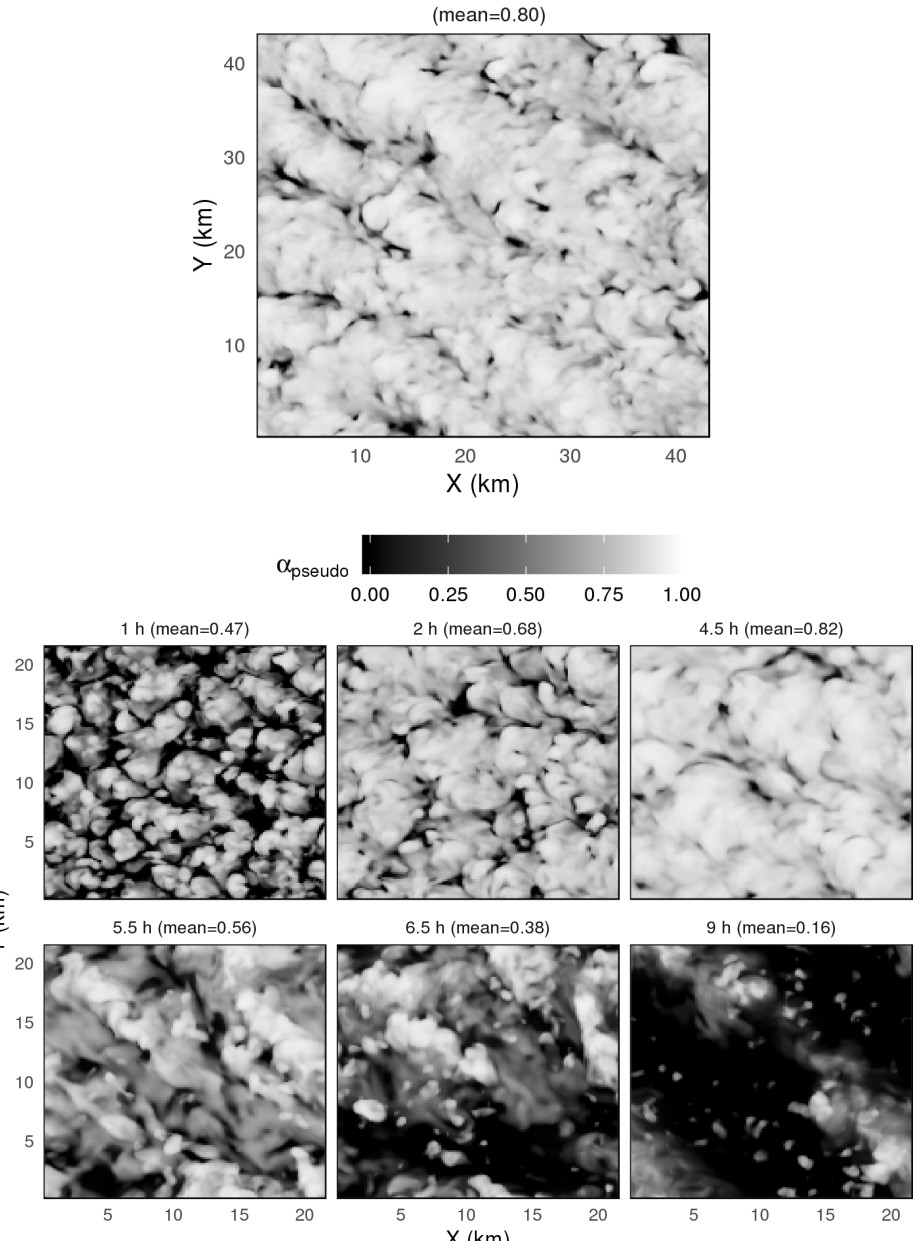

**Figure 4.** Pseudo-albedo of *ice0* at 4.5 h simulated time, shown for a domain four times larger in area than the baseline experimental setup (top) and at various times (indicated by panel titles) for the baseline setup (bottom).

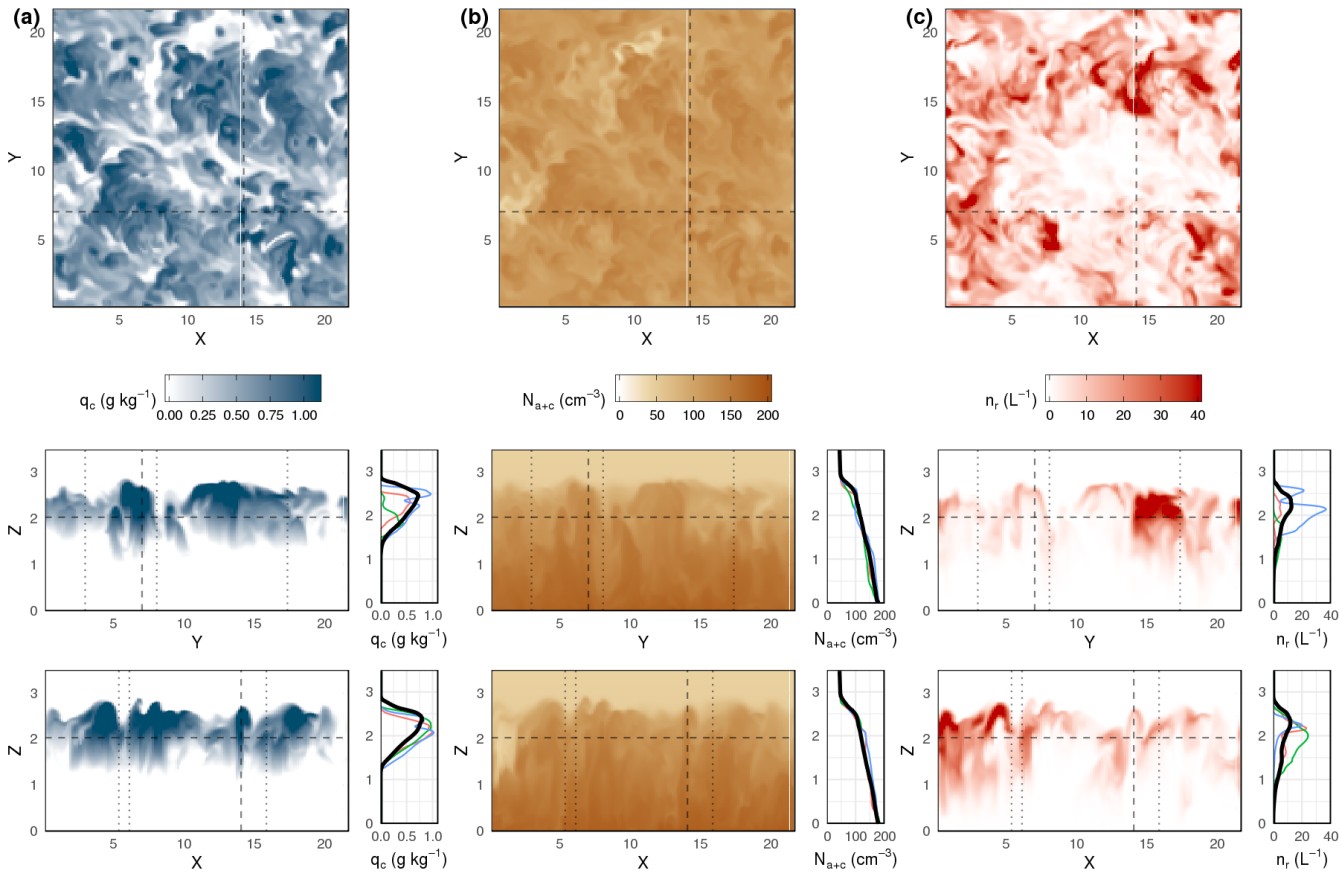

**Figure 5.** Horizontal and vertical transects (stacked vertically, coherent shading) of following variables (from left to right) of experiment *ice0* at time 4.5 h: cloud water mixing ratio, activated plus interstitial aerosol concentration, and raindrop concentration. Shading resolves 5[th] to 99[th] percentiles (capping values beyond the plotted range). Long-dashed lines mark transect locations. Short-dashed lines in vertical transects mark selected profiles that are shown in colors in the plot to their right; the black curve shows respective transect- average.

Figure 5 presents cross-sections of $q_c$, activated and unactivated aerosol $N_{a+c}$, and raindrop number concentration $N_r$ of the three-dimensional domain at onset of substantial rain. All quantities are marked by ample spatial heterogeneity. Compared against initial $N_{a+c}$ vertical profiles (see Section 2.2), lower altitudes within the PBL maintain their original values while upper portions experience a reduction by 50 cm$^{-3}$ (about 50 mg$^{-1}$), indicating weak PBL mixing.

In order to find out which processes dominate the temporal variability of $N_{a+c}$ in the PBL, Figure 6 provides a PBL-averaged budget of $N_{a+c}$ changes from specific sources and sinks. In the absence of ice, early $N_{a+c}$ reduction primarily results from entrainment of lesser FT aerosol concentrations, and together with collision-coalescence the PBL $N_{a+c}$ loss rate is steadily ~15 mg$^{-1}$ h$^{-1}$ over several hours prior to substantial rain (Fig. 6a). Once substantial rainwater builds up, drop-droplet collection removes aerosol at rapidly increasing rates that peak at ~50 mg$^{-1}$ h$^{-1}$. Evaporation of raindrops reintroduce CCN (one per drop) into the PBL but this rate is far outweighed by microphysical consumption (not shown).

Increasing the aerosol concentrations in the FT from 50 mg$^{-1}$ to initial concentrations in the PBL removes an early aerosol sink (shown as thicker, semi-transparent lines in Fig. 6a), offsetting some microphysical aerosol loss but only slightly delaying the onset of substantial rain. In this case, the FT acts as a relatively small CNN source (see yellow thick, semi-transparent line). Then, the PBL transitions comparable to lower FT levels (not shown).

To demonstrate the importance of collisional loss of activated aerosol for the cloud transition, we run another configuration in which aerosol concentration is fixed at 200 mg$^{-1}$, shown as "*ice0_no_loss*" in Figure 2. Resulting permanently high levels of $N_c$ (Fig. 2g) promote longer LWP growth (Fig. 2c) and delay the onset of substantial rain (Fig. 2e). Owing to lower RWP and reduced evaporation cooling and moistening below cloud, the PBL becomes stratified a bit more slowly (Fig. 2m) and remains somewhat better mixed (Fig. 2l); these effects would likely be greater if they were not offset by a substantial deepening of the PBL associated with longwave cooling (not shown) at a sustained cloud cover above 80 % (i.e., no breakup according to our definition) as seen in Fig. 2b. Switching off autoconversion results in a solid cloud deck with LWP plateauing at 1000 g m$^{-2}$ after 9 hours (not shown). As found by Yamaguchi et al. (2017) for subtropical warm stratocumulus, this case study illustrates the dependence of closed-to-open cell transitions on microphysical consumption of activated aerosol also in CAOs.

This study does not exhaustively examine all variables connected to PBL dynamics (e.g., Yang and Geerts, 2006). To understand if the timing of the breakup is related to the duirnal solar cycle, we switch off solar radiation. Even though cloud-top cooling intensifies by 10-20 %, we find insignificant responses when comparing against a simulated ensemble of the baseline setup (not shown).

## 3.2 Shortened overcast period in mixed-phase clouds

A priori, it is unclear how the addition of ice might change the aerosol-mediated overcast-to-broken cloud transition of CAOs (as elaborated in Section 1). Under the weakly to moderately supercooled conditions in this case (cloud top temperatures always warmer than ~−12° C; Fig. 2h), we expect primary ice formation to be dominated by immersion INP (e.g., de Boer et al., 2011). However, INP measurements are generally subject to order of magnitude uncertainties (e.g., Kanji et al., 2017), environmental INP spatiotemporal variability is greater than that (e.g., DeMott et al., 2010), and even greater uncertainties in predicted ice concentrations arise from incomplete understanding of secondary ice formation processes (e.g., Korolev et al., 2020; Korolev

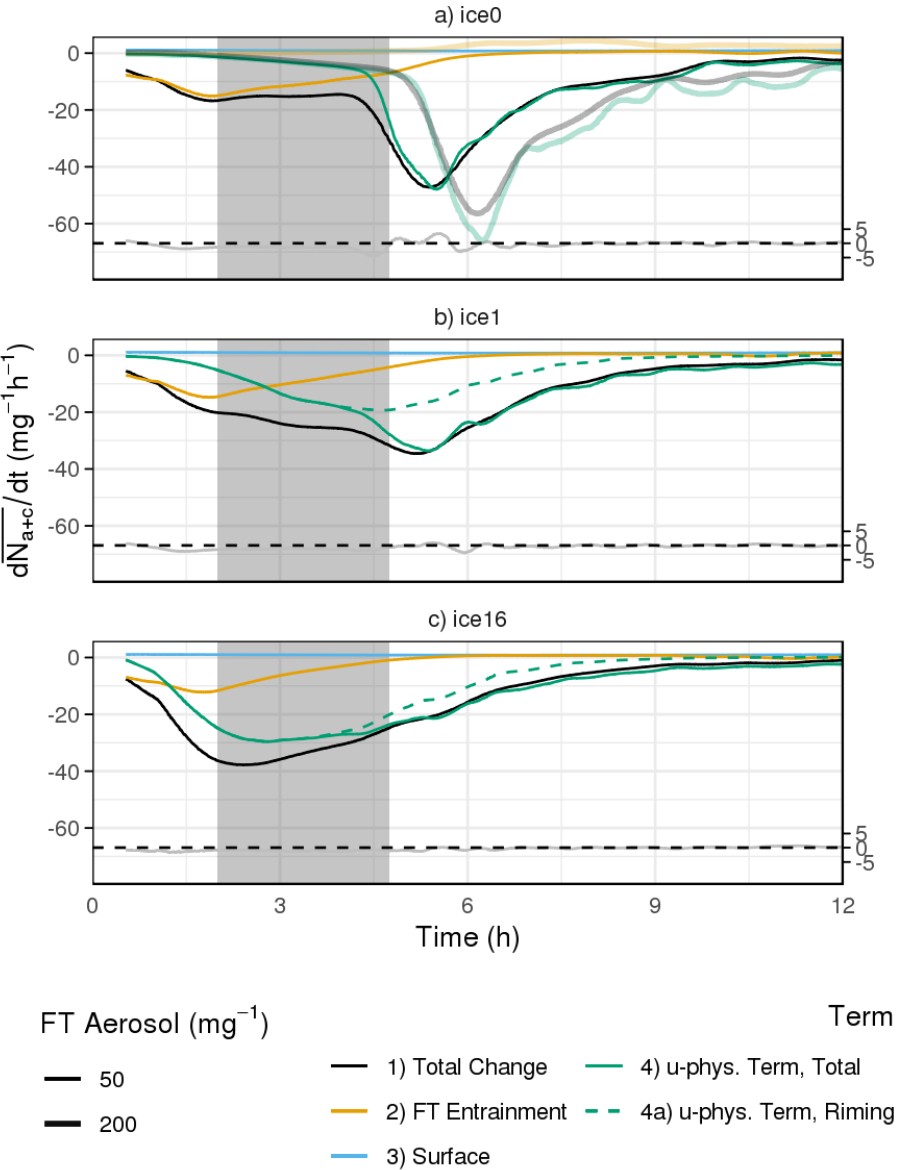

**Figure 6.** Budget of the temporal change of activated plus unactivated aerosol averaged over the PBL (actual change shown in black), and contributions from FT entrainment, surface source, and microphysical processes (in colors) for simulations with (a) *ice0*, (b) $N_{inp} = 1\ \mathrm{L}^{-1}$, and (c) $N_{inp} = 16\ \mathrm{L}^{-1}$. Thick, semi-transparent lines in panel (a) show an experiment with 200 $\mathrm{mg}^{-1}$ FT aerosol concentration, which matches the initial PBL concentration. Dashed green lines in panels (b) and (c) distinguish riming-related loss. Residuals of summed terms minus actual change are shown in grey (tied to right y-axis); zero difference is marked by dashed black line.

and Leisner, 2020). To provide a simple framework for approximately representing ice formation within cloud, we follow the Ovchinnikov et al. (2014) approach of specifying a fixed diagnostic immersion INP concentration, $N_{inp}$, which we vary in a sequence of simulations (see Section 2.2). In practice, this $N_{inp}$ is intended to represent the sum of highly uncertain primary and secondary ice formation processes.

We first fix $N_{inp} = 1 \, \text{L}^{-1}$ ("*ice1*") and compare with *ice0* to examine the role of modest ice in these transitions. With modest ice formation, we find almost no change to the onset time of substantial rain, but a subsequent overcast period that is shortened by about half (Fig. 2b). We next illustrate that three effects – acting prior to substantial rain onset and all connected to riming – cause an accelerated breakup downwind. We refer to following three effects collectively as *preconditioning by riming*. The addition of modest ice formation leads to the following changes:

(1) Reduction of the liquid water path (by ∼200 g m$^{-2}$ down to ∼400 g m$^{-2}$, Fig. 2c) from ice depositional growth and riming (shown in Fig. 8 as green dashed line and examined further below), leading to a peak ice water path ∼150 g m$^{-2}$ (Fig. 2d), mostly consisting of snow and to lesser extend of graupel and cloud ice (shown at the bottom of Fig. 7).

(2) More rapid reduction of $N_c$ (Fig. 2g) through intense riming (another collisional process that consumes activated aerosol; green dashed line in Fig. 6b) where both $q_c$ and ice water mixing ratio $q_i$ (including snow, graupel, and cloud ice) are at least moderate (examined further below in Fig. 7), leading to greater loss rates of activated aerosol before substantial rain onset (sustained up to ∼20 mg$^{-1}$ h$^{-1}$) compared to *ice0* (cf. Fig. 6a and 6b).

(3) Earlier precipitation (prior to the onset of substantial rain) in the form of riming-grown ice crystals (Fig. 2f at the surface and Fig. 7 at cloud base by precipitation type), foremost snow particles, that either sublimate directly or first melt and then evaporate below cloud leading to a moistening and cooling in these layers.

Figure 7 shows time-height plots of $q_c$, $q_i$, rainwater mixing ratio $q_r$ and overlaid microphysical aerosol loss rates for three experiments. In comparison to *ice0* (Fig. 7a), aerosol loss in *ice1* (Fig. 7b) strengthens earlier (between 2 and 4.5 h) and vertically overlaps with altitudes of moderate to high $q_c$ and moderate $q_i$ (between 1.5 and 2.5 km).

To highlight the mass-related impact of riming, Fig. 8 shows profiles of microphysical source terms of ice mixing ratio for selected times. In *ice1* (top panel) the transfer between water and ice phase ("Freezing Minus Melting", shown as solid green line) is the main source between 1.5 and 2.5 km and a major sink below 1 km altitude. Riming (shown as dashed green line) comprises effectively all direct transfer from water to ice phase. By comparison, riming is at least twice as strong in producing ice mass than depositional growth.

The combination of (1) and (2) leaves the onset time of substantial rain nearly unchanged (Fig. 2e) and produces similar precipitation rates (both liquid and frozen) at the surface shortly after substantial rain onset (Fig. 2f); reducing both LWP and $N_c$ results in comparable LWP/$N_c$ ratios across simulations that likely relate to comparable rates of cloud-base precipitation (Comstock et al., 2004, or here in the terms of rainwater paths) that then simultaneously exceed the 25 g m$^{-2}$ threshold. However, substantial rain contribution to $N_{a+c}$ consumption is delayed in *ice1* until nearly ∼4.5 h (Fig. 6b).

The combined effect of (2) and (3), however, contributes to stratification of the PBL before substantial rain onset (Fig. 2m). This stratification modulates the PBL dynamics early towards a convective state, indicated by a greater frequency of thin clouds

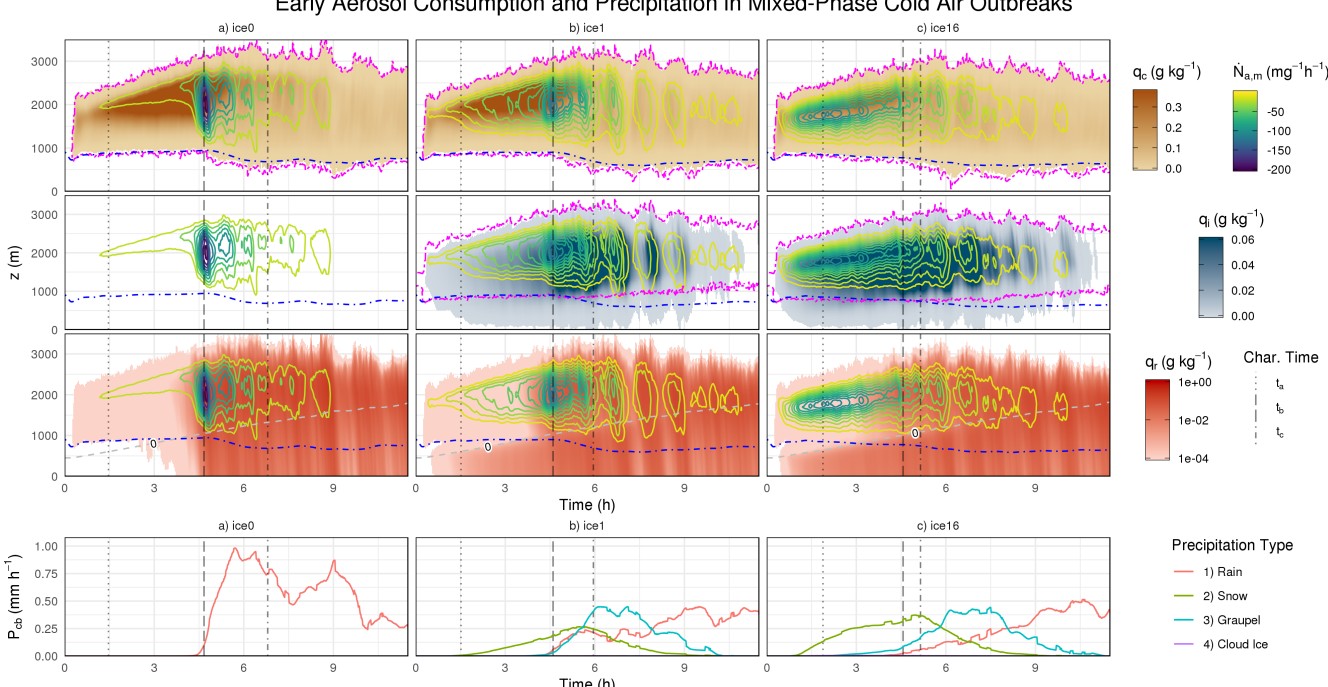

**Figure 7.** Time-height projections of horizontally averaged cloud water (top), cloud ice (second from the top), and rainwater mixing ratios (third from the top), shown for three simulations (by row). Overlaid in colored contours are aerosol consumption rates from microphysical collisions involving cloud droplets. Blue lines mark mean lifting condensation levels, using the lowest layer for calculations. Vertical lines highlight three characteristic times during PBL evolution. Magenta lines in cloud water and ice panels mark where layer-maximum supersaturation (with respect to liquid and ice, respectively) is zero. The bottom panels show cloud-base precipitation rates, $P_{cb}$, resolved by type (line color).

and the presence of few thick clouds already at substantial rain onset (cf. solid red lines in Fig. 3a and 3b, top panels) and also weakens in-cloud vertical winds earlier (Fig. 3a and 3b, bottom panels). Compared to *ice0*, the cloud deck breaks apart $\sim 1$ h sooner (Fig. 2b) by our definition. Using an alternative cloud cover threshold of 50 %, the breakup would be $\sim 2$ h sooner (Fig. 2b). At average horizontal PBL wind speeds, breakup by an hour earlier translates into a distance downwind of $\sim 75$ km covered with less reflective cloud decks.

The dropping LWP levels following the appearance of rimed snow at the surface qualitatively agrees with CAO observations in Young et al. (2016). The magnitude in LWP decline is also similar to that observed during transitioning CAOs by Shupe et al. (2008) and Abel et al. (2017), though the latter study had more than two orders of magnitude greater ice number concentrations at breakup (with local maxima exceeding $100\,\text{L}^{-1}$).

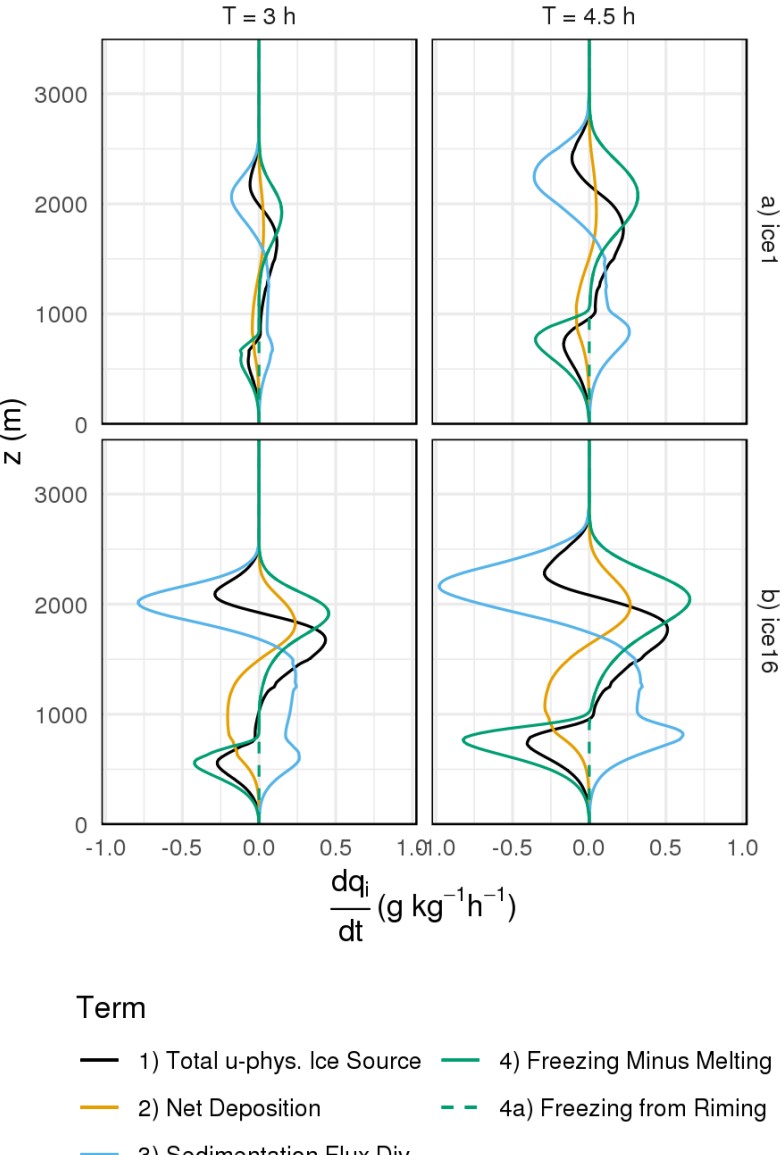

**Figure 8.** Profiles of ice mixing ratio instantaneous source terms at two selected times for *ice1* and *ice16* simulations. Black lines mark the overall net microphysical source and colored, solid lines resolve individual microphysical source terms: net deposition, sedimentation flux divergence, and freezing minus melting. The latter term is further refined to isolate riming (green dashed line), which effectively comprises all freezing by mass at supercooled temperatures.

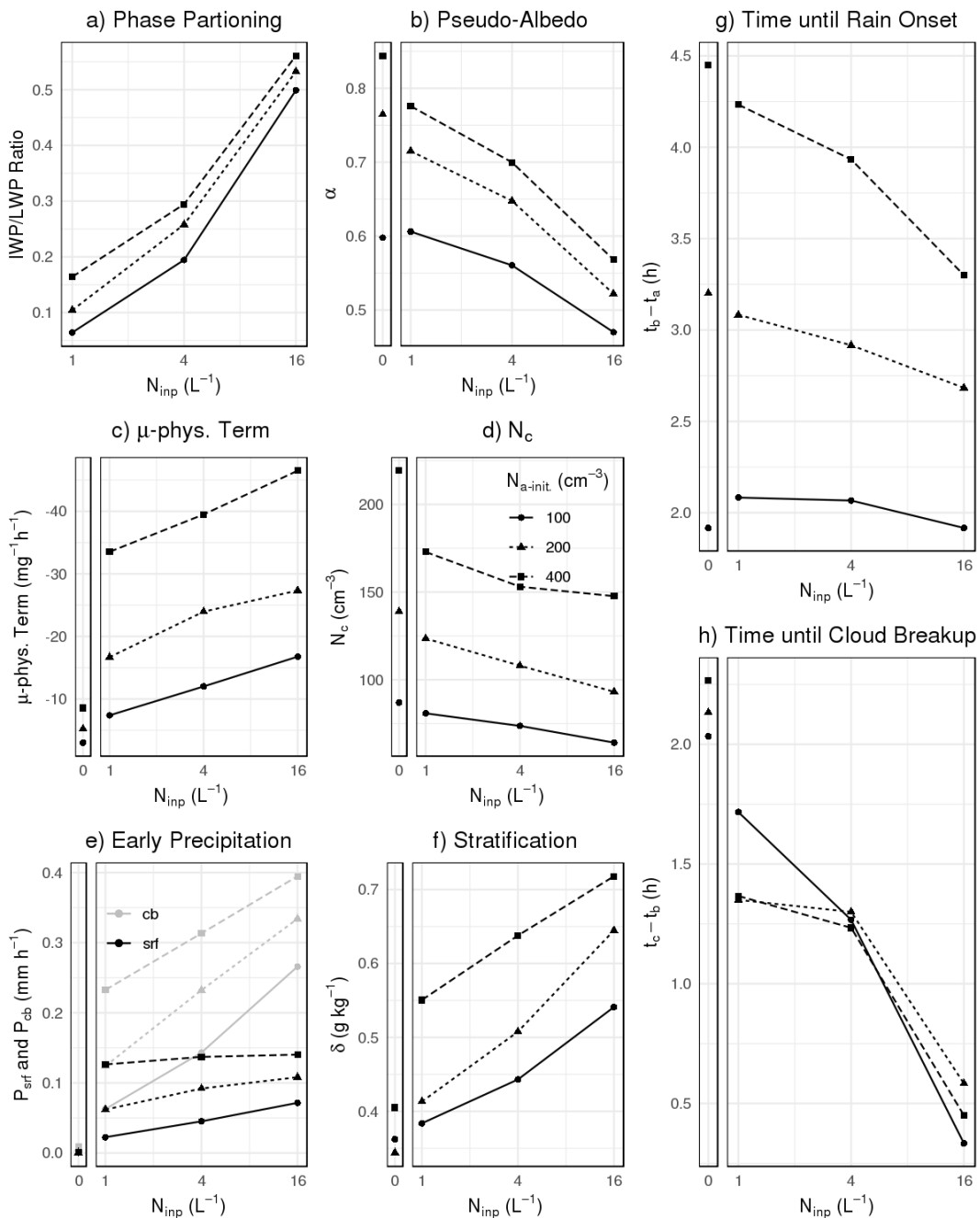

**Figure 9.** Dependence on INP concentrations (x-axis) and initial PBL $N_a$ (by line type and point shape) of metrics averaged over a 2-hour period prior to the onset of substantial rain $t_b$ (in this study termed *preconditioning by riming*): (a) IWP/LWP-ratio, (b) pseudo-albedo, (c) PBL-average aerosol consumption, (d) in-cloud droplet number concentration, (e) early (i.e., prior to onset of substantial rain) precipitation rates at cloud base and surface (labelled "cb" and "srf", respectively), and (f) PBL stratification $\delta$. Also the impact on transition speed defined as duration with overcast cloud: (g) time from first overcast cloud deck formation, $t_a$, to onset of substantial rain, $t_b$, and (h) time from substantial rain onset until cloud deck breakup, $t_c$.

## 3.3 Shortening of the overcast period in a concentration-dependent manner

Observations of ice concentrations at similar temperature ranges (e.g., 250-260 K in Young et al. 2016, 267-270 K in Huang et al. 2017, and 258-276K during a CAO transition in Abel et al. 2017) suggest a possible range spanning several orders of magnitude (1-10, 1-40, and 0.1-100 $L^{-1}$, respectively). To gauge the impact of greater concentrations, we quadrupled $N_{inp}$ to 4 $L^{-1}$ ("*ice4*") and again to 16 $L^{-1}$ ("*ice16*").

Figure 9 summarizes resulting responses averaged over the period termed preconditioning by riming (Fig. 9a-9f, referring to grey-shaded periods in Fig. 2 and 6) and captures the duration of the overcast state, separated into the period from the formation of overcast clouds until substantial rain onset (Fig. 9g) and the period from onset until the breakup of overcast clouds (Fig. 9h). Adding the diagnostics of Fig. 9g and 9h gives the total duration of the overcast state. In this subsection we analyze impacts for an initial PBL aerosol concentration of 200 $cm^{-3}$ (short-dashed line in Fig. 9).

Each successive $N_{inp}$ increase adds ~50 g $m^{-2}$ in IWP (Fig. 2d) while removing between 50-100 g $m^{-2}$ in LWP (Fig. 2c) during preconditioning by riming, magnitudes that generally correspond with responses in Arctic mixed-phase stratocumulus simulations by Stevens et al. (2018). Combined these raise the IWP/LWP ratio to up to ~0.5 for *ice16* (Fig. 9a), comparable to ratios reported from mixed-phase simulations of PBL clouds using high ice number concentrations (e.g., Eirund et al., 2019; Young et al., 2016). Overall diminished condensate (LWP plus IWP) and greater proportions of ice lead to a lower cloud albedo (Fig. 9b). Increasing $N_{inp}$ enhances riming (Fig. 6c and Fig. 9c), reduces $N_c$ (Fig. 9d), intensifying light precipitation at cloud base and to a lesser extent at the surface (Fig. 9e), and resulting in further stratification (Fig. 9f). While the onset of substantial rain is surprisingly insensitive to $N_{inp}$ (Fig. 9g), the time from rain onset to cloud breakup (Fig. 9h) is shortened from over 2 h in *ice0* down to ~0.5 h in *ice16*. At onset, the *ice16* simulation develops a cloud vertical structure foremost composed of geometrically thin portions and a few of large geometric extent, all in the presence of relatively weak in-cloud vertical winds (Fig. 3c, red lines). In *ice0* and *ice1*, such geometric extents and vertical winds were only associated with a convective state 1-2 h after substantial rain onset (Fig. 3a and 3b). However, the progression of increasing RWP modal values before, during and after onset of substantial rain in *ice16* remains similar to *ice0* and *ice1* simulations.

The accelerated breakup between *ice0* and *ice16* measures ~1.5 h (Fig. 9h) and translates into a ~120 km distance downwind that would reflect much more sunlight and emit less longwave radiation in the absence of cloud ice. Using a cloud cover >50 % to define the overcast state results in a similarly timed breakup across mixed-phase experiments (i.e., $N_{inp} > 0$ $L^{-1}$) regardless of $N_{inp}$ (Fig. 2b).

## 3.4 Intensification of riming preconditioning in high aerosol environments

Finally, we investigate the role of initial PBL aerosol concentrations to represent the substantial variability on the U.S. east coast with air mass origin (Sorooshian et al., 2019).

The main effect of greater $N_a$ is to delay the onset of substantial rain (Fig. 9g), while the time from onset to breakup is remarkably insensitive to aerosol concentrations (Fig. 9h), with the exception of a configuration using *ice1* and $N_{a-init} = 100$

mg$^{-1}$ that showed a slightly delayed breakup compared to greater $N_{a-init}$ and this delay is smaller than the ensemble spread of baseline simulations (i.e., ~0.5 h, Fig. 2b).

Additional time prior to onset of substantial rain allows the PBL to further deepen (not shown) and to promote greater
IWP/LWP ratios (Fig. 9a), presumably due to a greater geometric depth of supercooled cloud where $N_{inp}$ can be activated in these simulations. Owing to larger liquid and ice water paths (not shown) and also more numerous and smaller particles, cloud albedo is greater (Fig. 9b). More time prior to substantial rain onset further reduces aerosol concentrations through prolonged dilution via entrainment as well as microphysical consumption (Fig. 9c), leading to $N_c$ progressively diminished from initial concentrations (Fig. 9d, cf. *ice0* and mixed-phase simulations). We further find greater rates of light precipitation (Fig. 9e, the
larger geometric extents of supercooled cloud presumably facilitates more precipitating riming-grown frozen hydrometeors) over an extended period that leads to additional stratification (Fig. 9f). Positive feedbacks during substantial rain always quickly reduce $N_c$ below 50 mg$^{-1}$ (and below 25 mg$^{-1}$ after several hours, not shown), resulting in a low-CCN state in all experiments, which has been observed far downwind of CAOs (Wood et al., 2017). The time between substantial rain onset and breakup is relatively weakly sensitive to the factor of four range of initial $N_a$ considered here (Fig. 9h), with the exception of one
configuration that we pointed out earlier.

These experiments support the initially suggested mechanism of *preconditioning by riming* as driver of accelerated breakup. Ingredients for longer overcast periods (here the sum of the periods of cloud formation to substantial precipitation onset, Fig. 9g, and from onset to cloud breakup, Fig. 9h) are lower ice nucleation particle concentrations and higher initial aerosol concentrations in both PBL and FT.

## 4  Discussion

Cold air outbreaks produce PBL clouds that undergo radiatively important transitions from stratiform, overcast to convective, broken cloud fields. Here we demonstrate that frozen hydrometeors accelerate these transitions in an $N_{inp}$-dependent manner mainly through riming-related responses that act prior to transition-inducing rain.

Morrison et al. (2012) highlights stabilizing mechanisms of ice in mixed-phase clouds. Even though we showed that (more)
ice leads to (progressively) faster breakups in cold air outbreaks, one stabilizing mechanism does emerge from our study: with more ice the PBL deepens less. A shallower PBL prevents portions of the clouds from being supercooled and from developing greater IWP over extended times that would accelerate breakups further. While our simulations assume a set $N_{inp}$, ice formation has been observed to intensify with lower temperatures (that a shallower PBL would miss), potentially amplifying this stabilizing mechanism. Future efforts should investigate why some properties, such as early precipitation and also early
cloud microphysical composition, scale linearly with $\log(N_{inp})$ while others, such as breakup timing, respond less regularly.

Ice number particle concentration, treated diagnostically here, should be considered prognostically in the near future by for example including realistic INP spectra and established multiplication processes, such as ice multiplication in connection to drizzle (e.g., Rangno and Hobbs, 2001) that in itself is uncertain as seen in model intercomparisons (e.g., Klein et al., 2009; de Roode et al., 2019). In situ measurements are required to provide better bounds to INP sinks and sources.

This study, like others before (e.g., Abel et al., 2017; Field et al., 2014), demonstrates that cold air outbreaks are complex systems. An evaluation of general circulation model's column physics should focus on the ability to capture positive feedbacks between precipitation and droplet numbers (necessitating a prognostic treatment of aerosol) as well as the progressive stratification that both appear as cornerstones for closed-to-open cloud transitions. Such evaluations have been done in the past for SCT (e.g., Neggers, 2015; de Roode et al., 2019), but lacked prognostic aerosol.

This study demonstrates that cold air outbreaks exhibit both shortwave and longwave cloud radiative effects. Between a simulation setup that transitions toward the broken cloud state, such as *ice0*, and one that remains overcast, for example *ice0_no_loss*, we find a pseudo-albedo difference of about 0.4 (Fig. 2i). Using a global, diurnal average solar insolation of 340 W m$^{-2}$, the shortwave effect translates into roughly 140 W m$^{-2}$. On the other hand, results vary in outgoing longwave radiation (Fig. 2j), which responds to changes in cloud-top temperature, cloud cover, and underlying surface temperature, counteracting the shortwave radiative effect. The longwave effects can roughly be approximated from the difference between *ice0* and *ice0_no_loss* at 12 h, about 40 W m$^{-2}$, leaving a total cloud radiative effect of about 100 W m$^{-2}$.

Meteorological factors and their dynamic range may exert more leverage than the microphysical controls examined here. We are currently considering an ensemble of cold air outbreaks and their transition speeds in response to varied forcings including large-scale meteorology to investigate driving factors under a range of conditions. Among these factors is the initial profiles 360 of moisture and stability. Even within the same CAO, differences in transition speed can occur as seen across neighboring trajectories in Fig. 1. For instance, breakup is faster along adjacent trajectories to the northeast, where subsidence is weaker according to the reanalysis.

Despite other governing factors, we expect that the same microphysical mechanisms should be at play in CAOs of different intensity or in different regions. Likewise we expect that the sensitivities shown here would generally hold for a differing 365 meteorological baseline, such as could be associated with a more or less rapid breakup compared to our selected trajectory. Few observational case studies exist. Relative to an observed CAO transition in the North Sea by Abel et al. (2017), we find a comparable evolution in micro- and macro-physical liquid cloud properties. For example, coincident remote sensing data indicated peak LWP beyond 400 g m$^{-2}$ before the transition, similar to mixed-phase simulations in this study. Even though Abel et al. (2017) found much a higher ice loading in the their final stages of the breakup, in-situ probes indicated 370 rimed particles before and after the transition, similar to cold air outbreaks in the Beaufort Sea during M-PACE (Fridlind and Ackerman, 2018) or in post-frontal open cellular clouds in the Southern Ocean (Huang et al., 2017). Preliminary measurements during ACTIVATE corroborate the common presence of rimed ice particles (pers. comm. Simon Kirschler and Christiane Voigt). Abel et al. (2017) further observed a similar intensification of the PBL moisture stratification from 0.3 to 1.5 g kg$^{-1}$ (shown in Fig. 2m) over the course of the transition. Lastly, preliminary size distributions during ACTIVATE (pers. comm. Luke 375 Ziemba, Richard Moore) indicate that there are often fewer CCN in the FT than in the PBL during CAOs, similar to Abel et al. (2017). Inspection of sequential geostationary images along the trajectory simulated for this case (not shown) suggests that an overcast state was sustained hours longer than our simulations that include ice. As discussed above, duration of the overcast state is sensitive to the choice of trajectory, and uncertainty in meteorological forcings remains uninvestigated. Furthermore, the microphysical sensitivity to accumulation mode aerosol could explain such a difference; we demonstrated that higher aerosol

concentrations available for CCN activation delay the cloud transition. Lastly, preliminary aerosol size distribution gathered during ACTIVATE indicate an abundance of small aerosol particles (pers. comm. Luke Ziemba, Richard Moore). An Aitken mode was not included in our simulations and activation of small particles during elevated at high supersaturations (found in our simulation in the presence of rain, not shown) might further delay the cloud breakup. However, no in-situ measurements are available to indicate plausibility of such a setup for this case.

The ongoing ACTIVATE campaign (Sorooshian et al., 2019) should allow for an observationally inferred assessment of riming. In the spirit of Popper's falsifiability (Popper, 2002) we should be looking for counterexamples in the observations, such as abundant riming without significant reduction in total aerosol (contrasting with CCN loss zones in our simulations, not shown here) or else the presence of ice in a high-LWC environment that failed to rime, ideally using in situ imaging probes that characterize both ice and supercooled liquid. Finding such contradictory observations could point towards an inflated

representation of this effect using our model parameterizations and setup.

Mixed-phase PBL clouds are thought to constitute a reason for the large spread in equilibrium climate sensitivity across general circulation models (e.g., Zelinka et al., 2020; McCoy et al., 2020). In a warming climate, in which PBL conditions for ice formations shift progressively into higher altitudes and latitudes, cold air outbreaks might be expected to produce less cloud ice and, thereby, break up slower, increasing cloud cover and solar reflectivity (as seen in Fig. 2i) and reducing longwave

emission (Fig. 2j). We argue that riming represents a potentially important confinement of a negative cloud-climate feedback, assuming that ice formation will be generally weaker with increasing temperature.

## 5 Conclusions

CAOs strongly modulate the local albedo by forming overcast, stratiform cloud decks that transition into broken, convective cloud fields downwind. This study investigates the role of mixed-phase processes in these transitions. We show that, regardless

of cloud characteristics, transitions are triggered by the onset of substantial rain and mediated by rapidly depleting PBL CCN. The presence of frozen hydrometeors, mostly in the form of snow, accelerates transitions through the following riming-related effects prior to substantial rain onset: (1) reduction of cloud liquid water, (2) faster depletion of PBL CCN, and (3) early, light precipitation of riming-grown snow particles that stratify the PBL through sublimation or melting and evaporation. These effects, collectively termed *preconditioning by riming*, leave the timing of substantial rain onset unaffected while modulating

the PBL early towards the convective state. Preconditioning by riming scales with specified diagnostic INP concentration that serves as a proxy for uncertain ice formation strength, leading to progressively shorter overcast states. Increasing initial PBL CCN delays the onset of precipitation but owing to prolonged microphysical CCN loss prior to the onset, the transitions towards the broken state is largely indifferent to initial CCN concentration. The use of a CAO case in the NW Atlantic sets the stage for future ACTIVATE observations to assess the plausibility of this potential negative cloud-climate feedback with its roots in

mixed-phase microphysics.

*Author contributions.* FT, AA, and AF designed the simulations and FT carried them out. FT performed the formal analysis and visualized the data. FT prepared the initial draft and AA and AF reviewed and edited the manuscript.

*Competing interests.* The authors declare that they have no conflict of interest.

*Acknowledgements.* We would like to thank George Tselioudis, Brian Cairns, and Paquita Zuidema for insightful discussion as well as
NASA Advanced Supercomputing (NAS) for computing time. We thank two anonymous reviewers for their careful reading and insightful comments, which have improved the manuscript. This work is part of the ACTIVATE Earth Venture Suborbital-3 (EVS-3) investigation founded by NASA's Earth Science Division and managed through the Earth System Science Pathfinder Program Office, also supported by funding from the NASA Modeling, Analysis, and Prediction Program. We acknowledge the use of imagery from the NASA Worldview application (https://worldview.earthdata.nasa.gov/), part of the NASA Earth Observing System Data and Information System (EOSDIS).

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
