# Peer review of "Preconditioning of overcast-to-broken cloud transitions by riming in marine cold air outbreaks"

_Atmospheric Chemistry and Physics, 2021_

## Referee Comment (RC1)

The study titled "**Preconditioning of overcast-to-broken cloud transitions by riming in marine cold air outbreaks**" by Tornow et al. investigates the impact of frozen hydrometeors on an overcast-to-broken cloud transition during a marine cold air outbreak (CAO). The study makes use of a series of LES, which were set up following the ACTIVATE campaign in the NW Atlantic. The authors find that the formation of precipitation is necessary in order to simulate a cloud deck breakup, which is further accelerated if the ice nucleating particle concentration is increased. In this case, rimingrelated processes can trigger this accelerated breakup and precondition the cloud deck to transition from an overcast-to-broken cloud field.

The manuscript is well-written and contains very interesting findings which are presented in comprehensive figures and are described very clearly. The study adds to our current understanding of mixed-phase cloud (MPC) dynamics in CAOs and highlights the importance of microphysical processes, such as riming, for cloud field transitions and hence cloud albedo. Due to remaining lacks in our current MPC understanding and especially the importance of cloud field transitions for regional climate, I encourage publication in ACP. However, I have a few points that should be addressed prior to final publication.

**General comment**

1. In my opinion, the study would benefit from an analysis of the overall riming rates in the different simulations. The preconditioning by riming is an essential new finding of the manuscript, which could be stronger highlighted and supported. In Figure 6, the contribution of riming to aerosol concentrations is shown, but apart from that there is no graphical evidence of riming in the simulations. Thus, if available, riming rates and/or precipitation divided up into ice, snow and graupel (which would in turn allow an assessment of riming strength if the snow content is high) would help to make the main statement of the manuscript clearer and support several sections of the manuscript which mention e.g. "precipitation of riming-grown snow particles" (line 360).

**Specific comments**

**Abstract**

Line 11: I assume you mean "Greater boundary layer aerosol concentrations available for CCN activation" or "CCN concentrations" (i.e. in contrast to INP concentrations)?

Same line: "low-aerosol concentration" similar comment as above, I think it would be clearer to clearly distinguish between aerosols available for CCN activation and INPs.

Line 12 ff: The statement regarding impacts and the associated negative cloud-climate feedbacks seems slightly out of place here to me, as any affect of the cloud transition on climate parameters is not mentioned before. If you mention the climate effect of the cloud deck transition here, the effect of the increased ice and an abbreviated overcast state on albedo should be mentioned beforehand, e.g. in line 7. Otherwise I would suggest to move this section to the discussion.

**Introduction**

Line 18-24: Please provide some references to this passage.

Line 69: This research question itself sounds very similar to what was already answered by Eirund et al. (2019). I assume the difference is that you simulate a CAO in a Lagrangian perspective, while

simulations by Eirund et al. (2019) were idealized and stationary? It would be worth pointing this out here.

Line 72-73: This statement sounds as if your analyses were performed for a variety of CAO throughout the campaign. However, in section 2.1 you describe that you simulate one specific CAO during the shoulder season. In order to allow for an evaluation of the generality of the results found in this study as you mention here, it could be useful to expand your findings to a variety of CAOs, potentially in the discussion. A potential discussion point could be if your results would remain valid if the CAO index was different, e.g. the median of the collected indices?

**Simulations of a Cold Air Outbreak**

Fig 1: From the coastlines, it looks Figure 1a and b do not exactly cover the same area. It would be helpful to adapt either Figure 1a or Figure 1b, such that the cloud field and the MERRA-2 trajectories match up.

Lines 83-85: This sentence is a bit hard to read, maybe spilt into two sentences.

Line 89: Similar to what I mentioned above, I assume here you mean aerosols available for CCN activation?

Line 95: Please remove "01 May" after the Morrison and Grabowski reference.

Line 109: Is it justified to follow Abel et al. (2017) here, even though their case was in a different location and a different season?

**Results**

Line 137: Why is this threshold arbitrary (line 190) and not e.g. based on a percentile of the simulated cloud cover? A pdf as shown in Figure 3 of cloud cover could maybe show if the 75% threshold is reasonable considering the cloud field evolution.

Line 147: How are the ensemble simulations set up and why were they performed only for the ice0 case?

Line 159: Does the prognostic CCN implementation allow for recycling of CCN? If yes – doesn't the evaporation of rain below cloud release CCN, which could be re-entrained into the cloud layer?

Line 170: Abel et al. (2017) as well as Eirund et al. (2019) also show that precipitation formation is necessary for a cloud deck breakup, which might be worth noting as the latter studies also investigated MPCs.

Figure 2: You performed a "no aerosol loss" simulation, but did you also test the development of the cloud field under a scenario where autoconversion is not allowed as a baseline simulation (similar to Eirund et al. 2019 and Abel et al., 2017)? In their studies, the cloud deck remained completely overcast in the absence of precipitation (see my previous comment) - a similar experiment could strengthen your conclusion that precipitation formation is essential for cloud deck breakup also in this case.

Figure 5: It looks like the x-axes do not cover the full range of the vertical cloud water mixing ratio as well as the rain drop concentration shown in the small plots to the right of the contour plots.

Line 210: "substantial deepening of the PBL associated with longwave cooling" – do you have evidence of high LW cooling?

Line 214: "diurnal" instead of "duirnal"

Line 218: Is that really so unclear? It has previously been shown that cloud ice generally increases precipitation (Knight et al., 2002, Field & Heymsfield, 2015), which can then initiate regime transitions and cloud dissipation (Abel et al., 2017, Eirund et al., 2019).

Line 233: "ice vapor growth" – do you mean growth by deposition?

Same line: related to my comment 1, it would be very helpful to include the riming rates or the snow content/snow particle concentration e.g. in Figure 2 in order to follow this thought. Otherwise, the LWP reduction through riming sounds more like a suspicion rather than a fact.

Figure 6c: I assume, in the legend, the u-phys term should be dashed?

Line 240: similar to my above comment, where is the evidence for precipitation in the form of riminggrown ice crystals? Figure 2f only shows precipitation.

Line 285: Did you also investigate differences in longwave radiation between the simulations? As the difference between SST and cloud top temperature is quite large (Figure 1b, Figure 2h), is would be interesting to see the effect of changes in longwave radiation versus the simulated change in albedo/shortwave radiation.

**Discussion**

Line 349: How (and why) do you assume  $N_{inp}$  to change in a warming climate? Would the change in cloud ice alone not be sufficient for a negative cloud-climate feedback in the future?

I also think in the context of climate impact, it would be worth to again highlight the strong difference in albedo (as shown in Figure 2i) between the different simulations in the Discussion.

**References**

Knight, C. A., Knight, N. C., Dye, J. E., & Toutenhoofd, V. (2002). The mechanism of precipitation formation in Northeastern Colorado cumulus: I. Observations of the precipitation itself. Journal Atmospheric Research, 31(8), 2142–2147. https://doi.org/10.1175/1520-0469(1974)031h2142:tmopfi2.0.co;2

Field, P. R., & Heymsfield, A. J. (2015). Importance of snow to global precipitation. Geophysical Research Letters, 42, 9512–9520. https:// doi.org/10.1002/2015GL065497

---

## Author Comment (AC1)

[Figure]

**Fig. S1**: Similar to Figure 2 in the manuscript (please see for details), we added here two ice0 simulations: "L40" refers to domain size of (~40km)$^2$, "no rain" refers to switched off autoconversion.

[Figure]

**Fig. S2**: Heating rate profiles at 4.5 h.

[Figure]

**Fig. S3**: Isolines of 50 % (left) and 75 % (right) cloud cover. From MYD06 data we determine cloud cover over regions of $(0.5°)^2$, comparable to the extent of our large domain simulation seen in Fig. 4a. A $(1 \text{ km})^2$ pixel is categorized as cloudy if the cloud optical depth is greater than or equal 2.5.

---

## Author Response (AR2)

**Reviewer 1**

**The manuscript is well-written and contains very interesting findings which are presented in comprehensive figures and are described very clearly. The study adds to our current understanding of mixed-phase cloud (MPC) dynamics in CAOs and highlights the importance of microphysical processes, such as riming, for cloud field transitions and hence cloud albedo. Due to remaining lacks in our current MPC understanding and especially the importance of cloud field transitions for regional climate, I encourage publication in ACP. However, I have a few points that should be addressed prior to final publication.**

We thank the reviewer for the careful reading and insightful comments, which have improved the manuscript as addressed point-by-point below. Line numbers refer to the marked-up manuscript and the figures that were revised are provided at the end of this document.

**General comment**

1. **In my opinion, the study would benefit from an analysis of the overall riming rates in the different simulations. The preconditioning by riming is an essential new finding of the manuscript, which could be stronger highlighted and supported. In Figure 6, the contribution of riming to aerosol concentration is shown, but apart from that there is no graphical evidence of riming in the simulations. Thus, if available, riming rates and/or precipitation divided up into ice, snow and graupel (which would in turn allow an assessment of riming strength if the snow content is high) would help to make the main statement of the manuscript clearer and support several sections of the manuscript which mention e.g. "precipitation of riming-grown snow particles" (line 360).**

Agreed, we have added cloud-base precipitation partitioned by type (showing that snow dominates early precipitation) to the bottom of Fig. 7 and changed the text accordingly:
ll. 258-259: "…(Fig. 2f at the surface and Fig. 7 at cloud base by precipitation type)."
ll. 252-253: "…(shown at the bottom of Fig. 7)."
Caption of Fig. 7 is extended: "*The bottom panels show cloud-base precipitation rates, $P_{cb}$, resolved by type (line color)."*

We have also added a new Fig. 8 showing a profile of ice mass microphysical source terms for two selected times and two setups. As now noted, the terms indicate that the overwhelming entirety of (direct) mass transfer from supercooled water to ice is attributed to riming. This plot is integrated into the manuscript as follows:
l. 251: "…(shown in Fig. 8 as green dashed line and examined further below)."
ll. 264-268: "*To highlight the mass-related impact of riming, Fig. 8 shows profiles of microphysical source terms of ice mixing ratio for selected times. In ice1 (top panel) the transfer between water and ice phase ("Freezing Minus Melting", shown as solid green line) is the main source between 1.5 and 2.5 km and a major sink below 1 km altitude. Riming (shown as dashed green line) comprises effectively all direct transfer from water to ice phase. By comparison, riming is at least twice as strong in producing ice mass than depositional growth."*

**Specific comments**

**Abstract**

**Line 11: I assume you mean "Greater boundary layer aerosol concentrations available for CCN activation" or "CCN concentrations" (i.e. in contrast to INP concentrations)?**

We have revised as suggested:
l. 11: "*…available as cloud condensation nuclei (CCN)…*"

**Same line: "low-aerosol concentration" similar comment as above, I think it would be clearer to clearly distinguish between aerosols available for CCN activation and INPs.**

We have revised as suggested:
l. 12: "*…low-CCN concentration…*"

**Line 12 ff: The statement regarding impacts and the associated negative cloud-climate feedbacks seems slightly out of place here to me, as any affect of the cloud transition on climate parameters is not mentioned before. If you mention the climate effect of the cloud deck transition here, the effect of the increased ice and an abbreviated overcast state on albedo should be mentioned beforehand, e.g. in line 7. Otherwise I would suggest to move this section to the discussion.**

We have added a short sentence to prepare the reader for the statement:
ll. 13-14: "*An ice-modulated cloud transition speed suggests the possibility of a negative cloud-climate feedback.*"

**Introduction**

**Line 18-24: Please provide some references to this passage.**

We have added references as suggested:
ll. 21-22: "*(e.g. Hartmann et al., 1992; L'Ecuyer et al., 2019)*"
ll. 23-24: "*(Kolstad et al., 2009; Fletcher et al., 2016)*"
l. 26: "*(Papritz et al., 2015; Papritz and Spengler, 2017)*"

**Line 69: This research question itself sounds very similar to what was already answered by Eirund et al. (2019). I assume the difference is that you simulate a CAO in a Lagrangian perspective, while simulations by Eirund et al. (2019) were idealized and stationary? It would be worth pointing this out here.**

We have added a footnote to page 3 that points out the two most significant differences: (1) the experiments in Eirund et al. (2019) were set up with zero average horizontal wind speed, while this study nudges to horizonal winds exceeding 15 m/s. In our simulations, such strong winds drive enormous surface fluxes that drive boundary layer deepening and that in turn affects the cloud deck evolution, including the morphological transition from closed roll convection to open cells. (2) Another major difference is that Eirund et al. (2019) did not consider (microphysical) depletion of CCN, which is responsible for the morphological transition from closed to open cells in our simulations and a focal point of our study. This footnote reads as follows:

*"[1]Eirund et al. (2019) examined idle Arctic stratocumuli and did not consider (microphysical) depletion of CCN, which is critical for the transition from closed to open cells in our simulations."*

**Line 72-73: This statement sounds as if your analyses were performed for a variety of CAO throughout the campaign. However, in section 2.1 you describe that you simulate one specific CAO during the shoulder season. In order to allow for an evaluation of the generality of the results found in this study as you mention here, it could be useful to expand your findings to a variety of CAOs, potentially in the discussion. A potential discussion point could be if your results would remain valid if the CAO index was different, e.g. the median of the collected indices?**

In ongoing work with other cases, we are finding similar mechanisms at play in cold air outbreaks of different intensity and in different regions, but we strongly agree that generality should be tested in future work. We have extended a paragraph to the Discussion:

ll. 368-392: "*Despite other governing factors, we expect that the same microphysical mechanisms should be at play in CAOs of different intensity or in different regions. Likewise, we expect that the sensitivities shown here would generally hold for a differing meteorological baseline, such as could be associated with a more or less rapid breakup compared to our selected trajectory. Few observational case studies exist. Relative to an observed CAO transition in the North Sea by Abel et al. (2017), we find a comparable evolution in micro- and macro-physical liquid cloud properties. For example, coincident remote sensing data indicated peak LWP beyond 400 g m$^{-2}$ before the transition, similar to mixed-phase simulations in this study. Even though Abel et al. (2017) found much a higher ice loading in their final stages of the breakup, in-situ probes indicated rimed particles before and after the transition, similar to cold air outbreaks in the Beaufort Sea during M-PACE (Fridlind and Ackerman, 2018) or in post-frontal open cellular clouds in the Southern Ocean (Huang et al., 2017). Preliminary measurements during ACTIVATE corroborate the common presence of rimed ice particles (pers. comm. Simon Kirschler and Christiane Voigt). Abel et al. (2017) further observed a similar intensification of the PBL moisture stratification from 0.3 to 1.5 g kg$^{-1}$ (shown in Fig. 2m) over the course of the transition. Lastly, preliminary size distributions during ACTIVATE (pers. comm. Luke Ziemba, Richard Moore) indicate that there are often fewer CCN in the FT than in the PBL during CAOs, similar to Abel et al. (2017). Inspection of sequential geostationary images along the trajectory simulated for this case (not shown) suggests that an overcast state was sustained hours longer than our simulations that include ice. As discussed above, duration of the overcast state is sensitive to the choice of trajectory, and uncertainty in meteorological forcings remains uninvestigated. Furthermore, the microphysical sensitivity to accumulation mode aerosol could explain such a difference; we demonstrated that higher aerosol concentrations available for CCN activation delay the cloud transition. Lastly, preliminary aerosol size distribution gathered during ACTIVATE indicate an abundance of small aerosol particles (pers. comm. Luke Ziemba, Richard Moore). An Aitken mode was not included in our simulations and activation of small particles during elevated at high supersaturations (found in our simulation in the presence of rain, not shown) might further delay the cloud breakup. However, no in-situ measurements are available to indicate plausibility of such a setup for this case.*"

We are currently completing a follow-up study that considers several well-observed cold air outbreaks and, as already stated (moved to ll. 362-367), examining the role of meteorological boundary conditions (including CAO index) that affect CAO cloud deck evolution more than, for example, $N_{inp}$. We expect to report additional findings soon.

**Simulations of a Cold Air Outbreak**

**Fig 1: From the coastlines, it looks Figure 1a and b do not exactly cover the same area. It would be helpful to adapt either Figure 1a or Figure 1b, such that the cloud field and the MERRA-2 trajectories match up.**

We have revised Fig. 1a as suggested.

**Lines 83-85: This sentence is a bit hard to read, maybe spilt into two sentences.**

We have revised as suggested:
l. 92: "*...which we detected values of up to ~10 K during 17th March. This maximum nears the 95th percentile...*"

**Line 89: Similar to what I mentioned above, I assume here you mean aerosols available for CCN activation?**

We have revised as suggested:
l. 98: "*...available for activation as CCN...*"

**Line 95: Please remove "01 May" after the Morrison and Grabowski reference.**

We have revised as suggested.

**Line 109: Is it justified to follow Abel et al. (2017) here, even though their case was in a different location and a different season?**

Preliminary ACTIVATE measurements from the NW Atlantic also indicate that FT aerosol concentrations are typically less than MBL concentrations, which we assume here for our baseline setup. We have extended the paragraph in the Discussion accordingly:
ll. 379-381: "*Lastly, preliminary size distributions during ACTIVATE (pers. comm. Luke Ziemba, Richard Moore) indicate that there are often fewer CCN in the FT than in the PBL during CAOs, similar to Abel et al. (2017).*"

**Results**

**Line 137: Why is this threshold arbitrary (line 190) and not e.g. based on a percentile of the simulated cloud cover? A pdf as shown in Figure 3 of cloud cover could maybe show if the 75% threshold is reasonable considering the cloud field evolution.**

As there is no universal definition for "cloud breakup", any cloud cover threshold, whether relative or absolute, is somewhat arbitrary. Sandu et al. (2010) used a relative change and defined breakups by a drop in cloud cover down to half the peak value. In this study such a definition would correspond to a 50% cloud cover threshold that we already consider (ll. 204-206, 277-280, 308-311) but now stress further in Section 3:
l. 151: "*Alternatively, we also consider a cloud cover threshold of 50 % (equivalent to Sandu et al., 2010).*"

We also now note that Christensen et al. (2020) used 75% to delineate overcast from broken cloud state:
ll. 150-151: "…,*here defined as cloud cover above 75 %, as in Christensen et al. (2020).*"

We have also added Fig. 1c to show how well this cloud cover threshold applies to MODIS products (from MYD06 data in which we define cloud cover as the fraction of pixels with a cloud optical thickness greater equal 2.5 – equivalent to LES diagnostic, though at a different spatial scale) and added short reference:
l. 152: "*MODIS data in Fig. 1c provides an impression of cloud cover.*"

**Line 147: How are the ensemble simulations set up and why were they performed only for the ice0 case?**

We ran an ensemble to crudely characterize uncertainty from turbulent noise, set up by varying the seed to the pseudo-random number generator applied to the initial fields of water vapor and potential temperature. We only run one ensemble because we assume the turbulent noise of *ice0* is representative of the other variants on the case. We have expanded the text in Section 2.2:
ll. 116-119: "*To obtain a crude characterization of uncertainty from turbulent noise, we run an ensemble of simulations for the baseline setup of ice0, which we effectively assume as representative of other setup variations. Here ensembles are run by varying the seed to the pseudo-random number generator applied to initial fields of water vapor and potential temperature.*"
and extended the caption of Figure 2: "*For ice0, we show the spread over an ensemble of three simulations obtained by changing the pseudo-random seed used in initialization of meteorological fields.*"

**Line 159: Does the prognostic CCN implementation allow for recycling of CCN? If yes – doesn't the evaporation of rain below cloud release CCN, which could be re-entrained into the cloud layer?**

Indeed, there is recirculation in that one CCN is released per evaporating raindrop, but one raindrop is the product of collisions among and with many cloud droplets, and thus that one CCN corresponds to a reduction in CCN numbers. We have added a short comment to Section 3.1:
ll. 215-216: "*Evaporation of raindrops reintroduce CCN (one per drop) into the PBL but this rate is far outweighed by microphysical consumption (not shown).*"

**Line 170: Abel et al. (2017) as well as Eirund et al. (2019) also show that precipitation formation is necessary for a cloud deck breakup, which might be worth noting as the latter studies also investigated MPCs.**

We have revised the statement as suggested:
ll. 184-186: "*The precipitation-induced breakup of overcast cloud deck is generally consistent with findings of Stevens et al. (1998), Savic-Jovcic and Stevens (2008) and Wang and Feingold (2009) in warm clouds and Abel et al. (2017) and Eirund et al. (2019) in mixed-phase clouds.*"

**Figure 2: You performed a "no aerosol loss" simulation, but did you also test the development of the cloud field under a scenario where autoconversion is not allowed as a baseline simulation (similar to Eirund et al. 2019 and Abel et al., 2017)? In their studies, the cloud deck remained completely overcast in the absence of precipitation (see my previous comment) - a similar**

**experiment could strengthen your conclusion that precipitation formation is essential for cloud deck breakup also in this case.**

In response to the reviewer's suggestion we have run the case without autoconversion, labeled "no rain" in the Figure S1. Without autoconversion the LWP increases until plateauing beyond 1000 g m$^{-2}$ and the cloud deck remains overcast throughout. We have added a sentence to Section 3.1:
ll. 227-228: "*Switching off autoconversion results in a solid cloud deck with LWP plateauing at 1000 g m$^{-2}$ after 9 hours (not shown).*"

**Figure 5: It looks like the x-axes do not cover the full range of the vertical cloud water mixing ratio as well as the rain drop concentration shown in the small plots to the right of the contour plots.**

We have revised the figure as suggested.

**Line 210: "substantial deepening of the PBL associated with longwave cooling" – do you have evidence of LW cooling?**

A profile of radiative heating rates is shown at 4.5 h (roughly associated with the peak LWP maximum) in Figure S2, showing the slight warming in the lower cloud and strong cooling near cloud top. It would be noteworthy if this expected dipole were not evident. We have added "(not shown)" after "longwave cooling" in the revised manuscript (l. 226):
l. 226: "*...with longwave cooling (not shown) ...*"

**Line 218: Is that really so unclear? It has previously been shown that cloud ice generally increases precipitation (Knight et al., 2002, Field & Heymsfield, 2015), which can then initiate regime transitions and cloud dissipation (Abel et al., 2017, Eirund et al., 2019).**

For clarity we have linked the sentence in question to the introduction:
l. 236: "*...(as elaborated in Section 1).*"
There we hypothesize a possible delay in cloud transitions as LWP reduction (from mixed-phase processes) may delay transition-initiating rain. We have added the two references mentioned (Knight et al., 1974, Field & Heymsfield, 2015) when discussing possible reasons for an accelerated transition:
ll. 73-74: "*A general increase in precipitation in the presence of ice as found in cumulus clouds (Knight et al., 1974) and across various cloud types (Field and Heymsfield, 2015) could be expected to further support a more rapid breakup.*"

We found it surprising that ice had so little impact on the timing of warm precipitation at sufficient rates to trigger transition, and we believe that this is owing to reduction of droplet number concentration associated with riming, which offsets the substantial decrease in LWP (that would otherwise presumably delay precipitation onset) as already stated in the manuscript (ll. 269-272).

**Line 233: "ice vapor growth" – do you mean growth by deposition?**

We have changed the phrasing to "ice depositional growth":
l. 251-252: *...ice depositional growth...*

**Same line: related to my comment 1, it would be very helpful to include the riming rates or the snow content/snow particle concentration e.g. in Figure 2 in order to follow this thought. Otherwise, the LWP reduction through riming sounds more like a suspicion rather than a fact.**

We have added a figure depicting ice mass microphysical budgets for selected times, which show that riming is by far the predominant mechanism by which water freezes, which is discussed in a new paragraph:

ll. 264-268: *"To highlight the mass-related impact of riming, Fig. 8 shows profiles of microphysical source terms of ice mixing ratio for selected times. In ice1 (top panel) the transfer between water and ice phase ("Freezing Minus Melting", shown as solid green line) is the main source between 1.5 and 2.5 km and a major sink below 1 km altitude. Riming (shown as dashed green line) comprises effectively all direct transfer from water to ice phase. By comparison, riming is at least twice as strong in producing ice mass than depositional growth."*

**Figure 6c: I assume, in the legend, the u-phys term should be dashed?**

We have revised the figure as suggested.

**Line 240: similar to my above comment, where is the evidence for precipitation in the form of riming-grown ice crystals? Figure 2f only shows precipitation.**

We have expanded Fig. 7 to show cloud-base precipitation by type and now note that snow is the dominant form of precipitation prior to cloud transition:

ll. 252-253: *"...(shown at the bottom of Fig. 7)."*

**Line 285: Did you also investigate differences in longwave radiation between the simulations? As the difference between SST and cloud top temperature is quite large (Figure 1b, Figure 2h), is would be interesting to see the effect of changes in longwave radiation versus the simulated change in albedo/shortwave radiation.**

Agreed, we have revised Fig. 2 to show upwelling longwave radiation at 5 km, the top of the domain (panel j). Indeed, the upwelling longwave drastically changes over time from changes in cloud-top height, cloud cover, and underlying surface temperature. We have added a back-of-the-envelope calculation that shows longwave cloud radiative effects offsetting a non-trivial fraction of the shortwave effects in a new paragraph in the Discussion section:

ll. 355-361: *"This study demonstrates that cold air outbreaks exhibit both shortwave and longwave cloud radiative effects. Between a simulation setup that transitions toward the broken cloud state, such as ice0, and one that remains overcast, for example ice0_no_loss, we find a pseudo-albedo difference of about 0.4 (Fig. 2i). Using a global, diurnal average solar insolation of 340 W m$^{-2}$, the shortwave effect translates into roughly 140 W m$^{-2}$. On the other hand, results vary in outgoing longwave radiation (Fig. 2j), which responds to changes in cloud-top temperature, cloud cover, and underlying surface temperature, counteracting the shortwave radiative effect. The longwave effects can roughly be approximated from the difference between ice0 and ice0_no_loss at 12 h, about 40 W m$^{-2}$, leaving a total cloud radiative effect of about 100 W m$^{-2}$."*

**Discussion**
**Line 349: How (and why) do you assume $Ni$ to change in a warming climate? Would the change in cloud ice alone not be sufficient for a negative cloud-climate feedback in the future?**

While one could indeed consider the temperature dependence of ice formation, we are simply referring to the very simple principal that a warmer boundary layer can be expected to have less ice, as we now note explicitly:

ll. 404-405: "*...assuming that ice formation will be generally weaker with increasing temperature.*"

**I also think in the context of climate impact, it would be worth to again highlight the strong difference in albedo (as shown in Figure 2i) between the different simulations in the Discussion.**

For our cloud radiative effect comparison mentioned in the point before last, we now include a back-of-the-envelope estimate of shortwave radiative effect of about 140 W m$^{-2}$:

ll. 355-358: "*Between a simulation setup that transitions toward the broken cloud state, such as ice0, and one that remains overcast, for example ice0_no_loss, we find a pseudo-albedo difference of about 0.4 (Fig. 2i). Using a global, diurnal average solar insolation of 340 W m$^{-2}$, the shortwave effect translates into roughly 140 W m$^{-2}$.*"

**Reviewer 2**

**I lost my comments before submitting the preview. When I hit the preview, they were not there anymore. I did not copy / paste this before hitting the preview, so it is lost. This is a quick, shorter retype, in a different state of mind of course, so apologies for the brevity. If I get to review this again, I will work in a separate app, cut/paste into this form, and avoid this data loss, so that review will be better. I am asking the Editor to warn other reviewers about this pitfall.**

**Tornow et al. investigate the impacts of riming on the transition from overcast to broken/open cloud fields during a Cold Air Outbreak using Lagrangian LES simulations. With simulations that have no ice, they demonstrate the importance of precipitation and loss of activated aerosol for the transition from overcast to broken clouds. Using simulations that include ice nuclei, they show that riming can lead to an acceleration of this transition through three different processes: (1) Reduction of cloud liquid water, (2) early consumption of cloud condensation nuclei, and (3) early and light precipitation cooling and moistening below cloud. The authors refer to this as preconditioning by riming.**

**The findings of this study are interesting and further the understanding of the cloud transition in cold air outbreaks. The writing of the manuscript is very concise and clear and the findings are displayed in well-chosen figures that are easy to understand. The authors account for uncertainties by varying various parameters. I'm really interested to which extend the described phenomena can be observed in the future because the accelerated transition has important implications for cloud-climate feedbacks. Overall, I have some general comments, however, I would not consider these major comments and would suggest this submission for publication after minor revisions.**

We very much thank the reviewer for the careful reading and helpful suggestions that have now improved the manuscript, especially despite technical difficulties. In our point-by-point responses line numbers refer to the marked-up manuscript, and the figures that were revised are provided at the end of this document.

**General Comments:**

1. **Lines 144 – 145: As mentioned later the selection of the 75 % cloud cover threshold for a broken cloud field is somewhat arbitrary. It might make sense to show a MODIS image and indicate what 75 % cloud cover looks like in that image. This might help to justify the selection of this threshold or might also lead to the selection of a different threshold. I would think that 75 % cloud cover might be a little too high for a broken cloud field in Cold Air Outbreaks.**

We agree with the reviewer that the cloud cover threshold is arbitrary. We now reference the study by Christensen et al. (2020), which also used an absolute threshold of 75%:
ll. 150-151: "..., *here defined as cloud cover above 75 %, as in Christensen et al. (2020).*"

Throughout the original manuscript we also considered 50%, which corresponds to the relative threshold for breakup used by Sandu et al., (2010) that we now also reference:
l. 151: "*Alternatively, we also consider a cloud cover threshold of 50 % (equivalent to Sandu et al., 2010).*"

Following the reviewer's recommendation, we have produced a MODIS-based cloud cover based on that used in the analysis of our simulations. From MYD06 data we determine cloud cover over regions of $(0.5°)^2$, comparable to the extent of our large domain simulation seen in Fig. 4a. We do note that MODIS product pixel size and the LES mesh differ (1 vs. 0.15 km) and further note that the COD threshold (here 2.5, originally from Bretherton et al. 1997) is itself is arbitrary. In Figure S3 we also show isolines for 50% (left) and 75% (right) cloud cover.

The 75% isoline appears to distinguish brighter cloud streets from dimmer, open cellular fields downwind. The north-south gradient (with shorter overcast durations further north, as discussed in the next point below) is also captured using this 75% cloud cover threshold. In contrast, 50% cloud cover appears less telling as most of the cold air outbreak has a cloud fraction greater than 50% by above described method.

We have revised Fig. 1a and added Fig. 1c to show our cloud-cover diagnosis of $(0.5°)^2$ regions.

2. **This study argues that the accelerated transition and ice-mediated reduction in albedo may have important implications for cloud-climate feedbacks, i.e. a negative feedback. This remains speculative and needs to be borne out by other modelling and observational studies. In fact, Fig. 1a shows evidence to the contrary. The young (short-fetch) cloud albedo is higher to the north, and much lower south of Cape Hatteras. Helical roll circulations probably are omnipresent along the coast in the convective BL, amassing small convective cells, but further north the ice crystals near cloud top bridge the streets, whereas south of Hatteras, in the absence of ice, the cloud edge is defined by water droplets, which remains closer to the parent updrafts, less likely to bridge the cloud street subsidence regions, hence lower albedo.**

We agree that our speculation requires further study, including the gathering and analysis of observational evidence. In-situ observations of cold air outbreaks unsurprisingly reveal rimed particles, and we have extended a paragraph in the Discussion accordingly:

ll. 374-378: "*Even though Abel et al. (2017) found much a higher ice loading in their final stages of the breakup, in-situ probes indicated rimed particles before and after the transition, similar to cold air outbreaks in the Beaufort Sea during M-PACE (Fridlind and Ackerman, 2018) or in post-frontal open cellular clouds in the Southern Ocean (Huang et al., 2017). Preliminary measurements during ACTIVATE corroborate the common presence of rimed ice particles (pers.comm. Simon Kirschler and Christiane Voigt).*"

The reviewer's observations hint at meteorological controls. This study only considers one trajectory, which corresponds to a single set of meteorological boundary conditions. We expect that similar mechanisms should be at play in cold air outbreaks of different intensity and in different regions, including the North Sea, as discussed further below. We have extended a paragraph to the Discussion accordingly:

ll. 368-392: "*Despite other governing factors, we expect that the same microphysical mechanisms should be at play in CAOs of different intensity or in different regions. Likewise, we expect that the sensitivities shown here would generally hold for a differing meteorological baseline, such as could be associated with a more or less rapid breakup compared to our selected trajectory. Few observational case studies exist. Relative to an observed CAO transition in the North Sea by Abel et al. (2017), we find a comparable evolution in micro- and macro-physical liquid cloud properties. For example, coincident remote sensing data indicated peak LWP beyond 400 g m$^{-2}$ before the transition, similar to mixed-phase simulations in this study. Even though Abel et al. (2017) found much a higher*"

*ice loading in their final stages of the breakup, in-situ probes indicated rimed particles before and after the transition, similar to cold air outbreaks in the Beaufort Sea during M-PACE (Fridlind and Ackerman, 2018) or in post-frontal open cellular clouds in the Southern Ocean (Huang et al., 2017). Preliminary measurements during ACTIVATE corroborate the common presence of rimed ice particles (pers. comm. Simon Kirschler and Christiane Voigt). Abel et al. (2017) further observed a similar intensification of the PBL moisture stratification from 0.3 to 1.5 g kg$^{-1}$ (shown in Fig. 2m) over the course of the transition. Lastly, preliminary size distributions during ACTIVATE (pers. comm. Luke Ziemba, Richard Moore) indicate that there are often fewer CCN in the FT than in the PBL during CAOs, similar to Abel et al. (2017). Inspection of sequential geostationary images along the trajectory simulated for this case (not shown) suggests that an overcast state was sustained hours longer than our simulations that include ice. As discussed above, duration of the overcast state is sensitive to the choice of trajectory, and uncertainty in meteorological forcings remains uninvestigated. Furthermore, the microphysical sensitivity to accumulation mode aerosol could explain such a difference; we demonstrated that higher aerosol concentrations available for CCN activation delay the cloud transition. Lastly, preliminary aerosol size distribution gathered during ACTIVATE indicate an abundance of small aerosol particles (pers. comm. Luke Ziemba, Richard Moore). An Aitken mode was not included in our simulations and activation of small particles during elevated at high supersaturations (found in our simulation in the presence of rain, not shown) might further delay the cloud breakup. However, no in-situ measurements are available to indicate plausibility of such a setup for this case."*

We are currently completing a follow-up study that considers several cold air outbreaks across differing regions and, as already stated (moved to ll. 362-367), examines the role of meteorological boundary conditions (including CAO index) that affect CAO cloud deck evolution more than, for example, $N_{inp}$. We expect to report additional findings soon.

3. **In the discussion section I would like to see some more comparison with observations. Is Abel et al. (2017) (mature marine post-frontal clouds) really the best choice here if it looks at a different location for CAOs? This study looks at a step-change environment, rapid air mass transformation. Very different in terms of aerosol supply and surface flux history, compared to Abel et al. The authors should probably at least add some more quantitative values from Abel et al. that can be compared and contrasted. Moreover, is it not problematic that the satellite imagery suggests that the "overcast state was sustained hours longer" than in the simulations when the maximum difference between all the simulations is only ~1.5 hours (Fig. 8h)? See comment 1 as well.**

To qualitatively compare with Abel et al. (2017), we now also note that their LWP maximum of about 400 g/m$^2$ appearing upwind of the cloud transition corresponds well with the mixed-phase simulations of this study:
ll. 372-374: "*For example, coincident remote sensing data indicated peak LWP beyond 400 g m$^{-2}$ before the transition, similar to mixed-phase simulations in this study.*"

To compare with Abel et al. (2017), we already included metrics showing the progressive PBL stratification during the transition (ll. 378-379). Also, as mentioned in our response to the reviewer's previous point, rimed particles are evident in the in-situ observation of Abel et al, (2017) and other cold air outbreak studies (e.g., Huang et al., 2017, Fridlind and Ackerman, 2018, and preliminary ACTIVATE observations).

Regarding different transition speeds in satellite observations versus our simulations, we note that the breakup speed sensitivity to ice nuclei concentrations actually varies by up to 2.5 h in the simulations. The duration of the overcast state results from adding the metrics in Fig. 8g (up to 1 h difference) and Fig. 8h (up to 1.5 h difference). We now articulate this additive aspect in Section 3.3:

l. 293: "*Adding the diagnostics of Fig. 9g and 9h gives the total duration of the overcast state.*"

If there were more accumulation mode aerosol in the PBL, we think that this microphysical sensitivity could partly explain the difference from the satellite-observed overcast state. As Section 3.4 already shows, higher aerosol concentrations available for CCN activation delay the cloud transition. Additionally, we now also discuss the possibility of activating smaller aerosol size modes than considered in this study, given the substantial peak supersaturations in the simulations. In principle, activation of smaller modes could delay cloud transitions. These smaller modes are evident in preliminary ACTIVATE measurements.

ll. 388-392: "*Lastly, preliminary aerosol size distribution gathered during ACTIVATE indicate an abundance of small aerosol particles (pers. comm. Luke Ziemba, Richard Moore). An Aitken mode was not included in our simulations and activation of small particles during elevated at high supersaturations (found in our simulation in the presence of rain, not shown) might further delay the cloud breakup. However, no in-situ measurements are available to indicate plausibility of such a setup for this case.*"

**Minor Comments:**

1. **Line 23: "capped by strong subsidence"**

We have revised as suggested.

2. **Section 2.1: I think it would be good to add some more description of how this specific CAO event was chosen. What observations (if any) are available for this CAO event?**

As now stated, for this pre-campaign study, we were motivated to consider a case in the NW Atlantic (we added this information to Section 2.1):

ll. 89-90: "*Location and timing of this case are favorable as they align with the ongoing ACTIVATE campaign.*"

We also now note that the case was selected on the basis of weather-state analysis of satellite imagery by George Tselioudis:

l. 88-89: "*This CAO constitutes a shoulder season event and was selected on the basis of weather-state analysis of satellite imagery (pers. comm. George Tselioudis).*"

3. **Line 147: The ensemble members are not mentioned until this point. The authors might want to add some description of the 3 ensemble members, and how and why they were chosen.**

We ran an ensemble to crudely characterize uncertainty from turbulent noise, set up by varying the seed to the pseudo-random number generator applied to the initial fields of water vapor mixing ratio and potential temperature. We only run one ensemble because we assume the turbulent noise of *ice0* is representative of the other variants on the case. We have added this information to Section 2.2:

ll. 116-119: "*To obtain a crude characterization of uncertainty from turbulent noise, we run an ensemble of simulations for the baseline setup of ice0, which we effectively assume as representative of other setup variations. Here ensembles are run by varying the seed to the pseudo-random number generator applied to initial fields of water vapor and potential temperature.*"
and to the caption of Figure 2: "*For ice0, we show the spread over an ensemble of three simulations obtained by changing the pseudo-random seed used in initialization of meteorological fields.*"

4. **Section 2.2: In my opinion, the authors could add a table summarizing the setup of their simulation (which schemes are used/horizontal and vertical grid, etc.), to make it easier for the reader to see the whole setup at one glance.**

We have added a table as suggested.

5. **Line 114: Do the authors potentially mean 230K? 130K seems excessively low.**

We used 130 K for the overlying isothermal layer in concert with overlying column-integrated water vapor and ozone to match the downwelling longwave radiation profiles that we computed from radiative computations using a much deeper vertical grid (up to 30 km). We added "isothermal" (l. 125) to clarify the setup:
l. 125: "*...overlying isothermal layer temperature...*"

6. **Does Figure 3 show the statistics of the whole domain or only where clouds are present?**

Statistics only include cloudy samples, as we now note in the Figure 3 caption.
Added to the caption of Fig. 3: "*...from cloudy columns within 3D domains...*"

7. **Please be consistent with the naming of "u-phys term" in Figure 6 and "u-phys loss" in Figure 8. Also add a legend for the dot dashed lines in Fig. 6a.**

We have revised Figures 6a and 8 accordingly.

8. **Overall, I like the content of all the figures and how it is displayed. However, I would improve some minor things in the figures. Here are my suggestions: in Fig. 3 and 6 I would put the legends outside the plot and make it larger like it is in Fig. 3. In the figures which have a colorbar (Fig. 4,5,7) I would improve the display of the colorbar, maybe put a black box around them and color the ticks in black instead of white. In Fig. 5 some of the plots have data going outside the range which should be corrected.**

We have revised Figures 4, 5, and 7 as suggested.

**Table and Figures relevant to the revision of**
**"Preconditioning of overcast-to-broken cloud transitions by riming in marine cold air outbreaks"**

**Table 1.** Baseline model setup.

[revised manuscript text omitted]

**Figure S1**: Similar to Figure 2 in the manuscript (please see for details), we added here two ice0 simulations: "L40" refers to domain size of (~40km)$^2$, "no rain" refers to switched off autoconversion.

[Figure]

**Figure S2**: Heating rate profiles at 4.5 h.

[Figure]

**Figure S3**: Isolines of 50 % (left) and 75 % (right) cloud cover. From MYD06 data we determine cloud cover over regions of $(0.5°)^2$, comparable to the extent of our large domain simulation seen in Fig. 4a. A $(1 \text{ km})^2$ pixel is categorized as cloudy if the cloud optical depth is greater than or equal 2.5.

---

## Author Response (AR3)

**Reviewer 1**

**General comment**

**The authors have substantially improved the manuscript and have included the reviewers' suggestions and comments. The manuscript especially benefitted from the expanded discussion session on the generality of the authors' findings and the inclusion of the precipitation contributions and ice mass microphysical source terms in Figures 7 and 8.**

**I have one small comment regarding the revised version of the manuscript, which I think should be addressed before final publication in ACP:**
**The authors added a footnote on page 3, pointing to differences between their work and work by Eirund et al. (2019). However, in my opinion these are quite significant differences that deserve more attention than a footnote. I think it is worth pointing out these differences in the main text (not as a footnote) to clearly distinguish this study from work by Eirund et al. (2019).**

We thank the Reviewer once more for carefully reading our manuscript and appreciate the renewed feedback. With respect to the Reviewer's comment, we have changed the manuscript as follows. We deleted the footnote and included following sentence at the end of Section 1 (ll. 82-84):
*"We note that our investigation differs from the Eirund et al. (2019) study of idle Arctic stratocumulus organization in two respects: (1) our meteorological context of a CAO in which not only are the mean winds not still, but gale-force, and (2) our focus on CCN depletion, which is critical to the downwind cloud transition here."*